# Off-policy Token clipped Supervised Fine-Tuning yields a robust cold-start

## Abstract

Supervised Fine-Tuning (SFT) is a critical step for adapting Large Language Models (LLMs) to specialized domains, often serving as a cold-start for subsequent reinforcement learning (RL). However, SFT's tendency to memorize a small set of expert data for a downstream task can impair generalization and lead to catastrophic forgetting of prior knowledge, undermining the promise of effective RL. In this paper, we demonstrate that this degradation primarily results from tokens in the expert data to which the base model assigns low probability. Specifically, we frame these as 'off-policy' tokens, as they represent a significant deviation from the model's current prior knowledge. Due to the nature of the log-likelihood objective, these off-policy tokens produce larger gradient magnitudes, destabilizing the training process. To investigate this phenomenon, we adopt a well-established clipping strategy, which is widely used to constrain per-token updates within a trust region. Applying this strategy to SFT moderates the learning process by constraining gradient updates from off-policy tokens, creating a more on-policy-like training dynamic. Through extensive experiments on the agentic benchmarks ALFWorld and ScienceWorld, we discover that this clipped approach, compared to standard SFT, reduces forgetting on out-of-distribution tasks by 11.54% and boosts final RL performance by 7.09%. Furthermore, latent-space analysis validates our initial claim, showing that applying the off-policy token clipped strategy results in less model's internal representational drift than standard SFT and is thus key to preserving prior knowledge.

## 1 Introduction

Large Language Models (LLMs) have found wide applications in complex reasoning and decision-making tasks (OpenAI et al., 2024; Team et al., 2025). Although LLMs are routinely pre-trained on billions of tokens, it is insufficient to produce models that are adept at specialized downstream tasks or capable of robust, multi-step reasoning (DeepSeek-AI et al., 2025; Liu et al., 2021). Therefore, post-training, which includes Supervised Fine-Tuning (SFT) and Reinforcement Learning (RL), is the critical stage for continually improving LLMs and enabling the acquisition of new abilities (Kumar et al., 2025; Zhang et al., 2024). Within this paradigm, SFT adapts a model to a new domain by training it on a curated set of expert data. This process directly instills task-specific behaviors and, critically, provides an essential cold-start for the subsequent RL phase (Ouyang et al., 2022; Qwen et al., 2025; Yu et al., 2025a; Wang et al., 2025).

Although this process is adept at cloning a specific behavioral policy (DeepSeek-AI et al., 2025; Wei et al., 2025), the model's tendency to memorize these static traces leads to impaired generalization and the catastrophic forgetting of pre-existing knowledge (Chu et al., 2025; Wu et al., 2025b; Shenfeld et al., 2025). This occurs as SFT inadvertently alters the model's internal representations, causing an erosion of the foundational knowledge acquired during pre-training. This degradation is particularly detrimental for the subsequent RL phase. A flawed cold-start means initializing the RL agent in a less generalizable and knowledgeable state, which limits the generation of useful exploratory experiences and imposes a ceiling on its performance (Huan et al., 2025; Zhao et al., 2025). Consequently, a fundamental question arises:

*What are the specific mechanisms within SFT that induce catastrophic forgetting, and can we mitigate them to yield a more robust cold-start?*

In response to this question, we investigate the training dynamics of SFT on the agentic benchmarks (Luo et al., 2025a; Shridhar et al., 2021), where an effective cold-start is essential (Shang et al., 2025). We find that the majority of representational damage occurs during the initial stages of fine-tuning. This period of high probability change on out-of-distribution tasks correlates directly with a suddenly elevated gradient norm, which we attribute to the LLM encountering tokens within the expert data that it assigns a very low probability to. This phenomenon directly results from the log-likelihood objective, which assigns disproportionately large gradient magnitudes to low-probability tokens, thereby biasing the learning process. We frame these low-probability tokens as 'off-policy' tokens, as they represent a significant deviation from the model's prior knowledge. This terminology originates from reinforcement learning, and we use it to draw an analogy: the data being learned from is not aligned with the model's current behavior. We therefore identify the gradient brought by these off-policy tokens as the primary mechanism behind the catastrophic forgetting of standard SFT. This insight suggests a possible strategy: to directly constrain the destabilizing gradient updates that originate from these off-policy tokens.

To investigate this, we adopt the clipping strategy widely used in trust region methods (Schulman et al., 2015; 2017), and develop a method we term *Off-Policy Token-Clipped SFT (OPC-SFT)*. This method computes a token-level probability ratio to measure the policy deviation induced by an update. To preserve the model's prior knowledge, the ratio is clipped for off-policy tokens, thus preventing the large gradient magnitudes they would otherwise cause. This mechanism directly tempers the influence of high-magnitude gradients generated by low-probability targets, preventing the destabilizing updates that cause knowledge degradation. We conduct extensive experiments to validate our claim. On the agentic benchmarks ALFWorld and ScienceWorld, OPC-SFT demonstrates substantial gains in generalization over conventional SFT, reducing out-of-distribution forgetting by 11.54% and boosting the final performance of a downstream RL agent by 7.09%. We support these findings with a latent-space analysis, showing OPC-SFT induces significantly less representational drift, and an analysis of probability dynamics, which demonstrates that it successfully clips drastic updates. Furthermore, we find that OPC-SFT is most pronounced when the initial gradient norm is large, which occurs when the expert data is substantially off-policy.

## 2 PRELIMINARIES

### 2.1 LLMs FINE-TUNING FRAMED AS AN RL PROBLEM

Let $\pi(y|x)$ denote the conditional generative distribution modeled by an LLM with parameters $\theta$. In generative reasoning tasks, the LLM sequentially generates an output sequence $y = (y_1, \ldots, y_T)$ by predicting one token at a time, given an input query prompt $x_0$. For complex tasks, this sequence $y$ often includes a chain-of-thought (CoT), verbalizing a step-by-step reasoning trace, followed by a final answer. From a reinforcement learning perspective, we can frame this sequential token-wise generation as a decision-making process. We define a state space $\mathcal{X}$ and an action space $\mathcal{Y}$. At each timestep $t$, the LLM serves as a policy $\pi : \mathcal{X} \rightarrow \Delta(\mathcal{Y})$, where $\Delta(\cdot)$ is the probability simplex. A state $x_t \in \mathcal{X}$ represents the prompt concatenated with all previously generated tokens, and an action $y_t \in \mathcal{A}$ corresponds to the next token to be generated. This token-wise generation process can be optimized either through supervised methods, where the policy is trained to mimic an expert sequence $y^*$, or by Reinforcement Learning with Verifiable Rewards (RLVR) methods, which leverage a reward function $R : \mathcal{X} \times \mathcal{Y} \rightarrow \{0, 1\}$ to guide the LLM towards desired behaviors.

### 2.2 SUPERVISED FINE-TUNING AND COLD-START

Downstream tasks in mathematics, coding, and agentic settings require capabilities that pre-training alone rarely provides. A brief cold-start phase therefore initializes the model with a small set of high-quality supervised demonstrations, transferring core skills such as multi-step reasoning and problem-solving format. This phase is implemented as supervised fine-tuning on a corpus $\mathcal{D} = \{(x, y^*)\}$, which minimizes the following objective:

$$\mathcal{L}_{\text{SFT}}(\theta) = \mathbb{E}_{(x,y^*)\sim\mathcal{D}}[-\log \pi_\theta(y^* \mid x)]. \tag{1}$$

SFT is an effective training paradigm that can rapidly improve performance; however, it is inherently off-policy because $\mathcal{D}$ is drawn from an expert distribution rather than from rollouts of the current LLM model. When trained on only a small set of new demonstrations with distributional shift,

the model can overfit, and this will limit generalization to scenarios not covered during training. Ultimately, this SFT phase yields an LLM model, $\pi_0$, referred to as the cold-start, which serves as the starting point for reinforcement learning.

## 2.3 CLIPPING STRATEGY IN POLICY OPTIMIZATION

Following the SFT cold-start, the initial LLM model $\pi_0$, hereafter also referred to as the policy model, is further optimized through a reinforcement learning phase. In this stage, the LLM learns from its own generated rollouts. For on-policy optimization, importance sampling is employed to estimate gradients using data collected by the old policy $\pi_{\theta_{\text{old}}}$. Trust region methods are applied to proximally control the update size and correct for distributional shift. Trust Region Policy Optimization (TRPO) achieved this by enforcing a hard constraint on the KL-Divergence between the current policy $\pi_\theta$ and $\pi_{\theta_{\text{old}}}$ (Schulman et al., 2015). However, this approach requires calculating vector products with the inverse Fisher Information Matrix, a second-order optimization procedure that is computationally prohibitive and numerically unstable for large-scale models. Consequently, Proximal Policy Optimization (PPO) (Schulman et al., 2017) is commonly employed as a clearer, more efficient alternative. PPO constrains the policy update by comparing the current policy $\pi_\theta$ to $\pi_{\theta_{\text{old}}}$. This is achieved using a probability ratio $r_t(\theta)$ and an advantage estimate $\hat{A}_t$ at timestep $t$:

$$r_t(\theta) = \frac{\pi_\theta(y_t \mid s_t)}{\pi_{\theta_{\text{old}}}(y_t \mid s_t)}. \tag{2}$$

This ratio is then used in a clipped objective function, which penalizes large deviations from the previous policy:

$$\mathcal{L}_{\text{CLIP}}(\theta) = \mathbb{E}_t \left[ \min\left( r_t(\theta)\hat{A}_t, \text{clip}(r_t(\theta), 1 - \epsilon, 1 + \epsilon)\,\hat{A}_t \right) \right]. \tag{3}$$

This proximal objective provides a simple yet effective on-policy correction that stabilizes training and improves generalization in agentic tasks.

## 3 OFF-POLICY TOKEN CLIPPED SUPERVISED FINE-TUNING

In this section, we provide an analysis of the SFT process on catastrophic forgetting. We begin by presenting an analysis that reveals a strong correlation between off-policy tokens and catastrophic forgetting, as measured by the model's probability change on out-of-distribution tasks. Based on this insight, we then adopt the clipping strategy on those off-policy tokens, a method termed OPC-SFT, to test the hypothesis that selectively constraining updates from these off-policy tokens mitigates forgetting. We ground our validation in the domain of agentic tasks, specifically using the agentic benchmark (Shridhar et al., 2021). This environment is an ideal testbed because the textual-based embodied task requires a cold-start for the LLM to learn the specific decision-making format, a capability usually absent during pre-training.

### 3.1 SFT PITFALLS: CATASTROPHIC FORGETTING AND OFF-POLICY TOKENS

Our investigation starts from the observation that SFT tends to reallocate probability mass toward task-specific patterns, often at the expense of general knowledge (Chu et al., 2025; Huan et al., 2025). Empirically, we post-train a warmed-up Llama3.2-3B-Instruct model on ALFWorld (Shridhar et al., 2021) with SFT and RL respectively, until they achieve comparable performance on the in-distribution test set. Then we evaluate its knowledge retention on a suite of out-of-distribution benchmarks including coding and QA tasks: GPQA (Rein et al., 2023), HumanEval (Chen et al., 2021), MBPP (Austin et al., 2021), and MMLU (Hendrycks et al., 2021a). We compare the probability change on these benchmarks for a model trained with SFT against one trained with RL. As shown in Fig. 1(a), the results demonstrate that SFT induces a significantly more drastic change in the model's probabilities than RL, meaning that it often achieves new-task gains by erasing prior knowledge. To determine when this knowledge degradation occurs, we analyze the progression of the probability change throughout the SFT process. As shown in Fig. 1(b), we plot the incremental probability change between consecutive training checkpoints. It reveals that the probability change induced during the first episode is substantially larger than in any subsequent episode. Specifically,

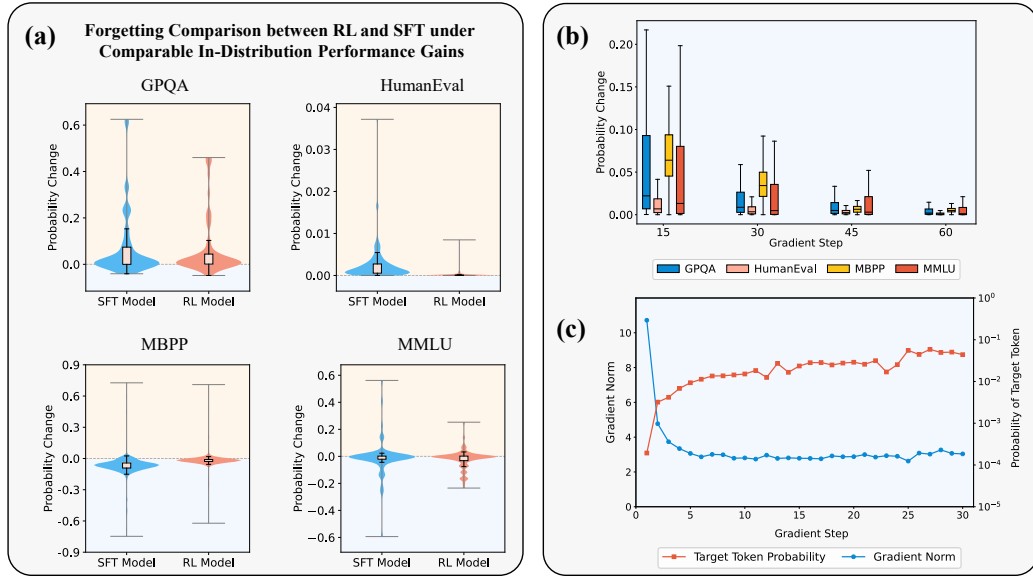

Figure 1: (a) Probability changes comparison between RL and SFT models after both achieving comparable in-distribution validation performance. (b) Probability changes during training. (c) Curves of gradient norm and target token probability for tokens in the bottom 1% quantile.

the incremental probability change observed on GPQA is 60.7% smaller in the second epoch compared to the first, while for MBPP, the reduction is 46.6%. This provides strong evidence that the majority of catastrophic forgetting happens during this initial stage. We further examine the evolution of target token probabilities and gradient norms during training. Fig. 1(c) shows that the high gradient norm drops abruptly during initial training, with target tokens simultaneously exhibiting the probabilities in the bottom 1% quantile increase suddenly. These high-magnitude gradients are, in turn, responsible for the large probability changes observed in Fig. 1(b). We identify this phenomenon, where the model encounters off-policy data that deviates from its prior knowledge, as a direct cause of catastrophic forgetting. In Sec. 3.2, we derive the relationship between gradient norms and target token probability by analyzing the SFT objective's gradient formulation.

## 3.2 THE PROBLEM OF LARGE GRADIENT NORM IN STANDARD SFT

The standard objective for SFT is to maximize the likelihood of an expert-provided response $y^*$ given the input query $x$. This is achieved by minimizing the negative log-likelihood loss for each sample in a dataset $\mathcal{D} = \{(x, y^*)\}$:

$$\nabla_\theta \mathcal{L}_{\text{SFT}}(\theta) = -\mathbb{E}_{(x,y^*)\sim\mathcal{D}} \left[ \frac{\nabla_\theta \pi_\theta(y^* \mid x)}{\pi_\theta(y^* \mid x)} \right]. \tag{4}$$

While this objective is intuitive, its training dynamics may exhibit sudden and excessively large gradient magnitudes. This phenomenon critically stems from the $\pi(y^*|x)$ term in the denominator. During the early cold-start phase, the model frequently assigns very low probabilities to expert targets, an event we empirically observed in Fig. 1(c). When this denominator approaches values as low as $10^{-4}$, the gradient's magnitude may become excessively large. While this behavior is not theoretically guaranteed, our fine-grained empirical study in Fig. 2 clearly reveals the phenomenon that low-probability tokens tend to produce much larger gradient norms. We partition tokens into ten probability bins. Tokens falling into the lowest-probability bin $[0, 0.1)$, which we term off-policy tokens in Sec. 3.1, exhibit an average gradient norm of 32.75, in sharp contrast to 2.02 for high-confidence tokens in the $[0.9, 1.0]$ range. This indicates that minimizing loss on off-policy tokens requires large parameter updates. Such aggressive updates overwrite previously learned policy behavior in order to fit unlearned data, making them a principal driver of catastrophic forgetting.

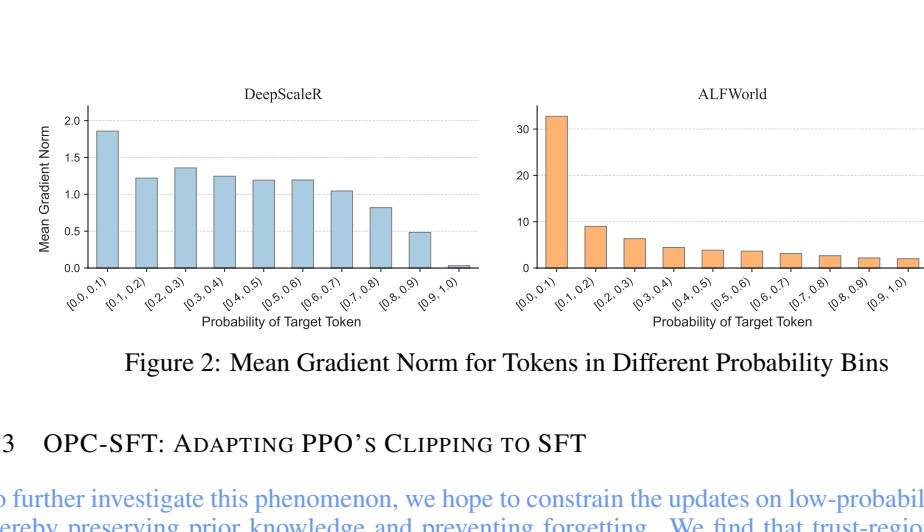

Figure 2: Mean Gradient Norm for Tokens in Different Probability Bins

### 3.3 OPC-SFT: ADAPTING PPO'S CLIPPING TO SFT

To further investigate this phenomenon, we hope to constrain the updates on low-probability tokens, thereby preserving prior knowledge and preventing forgetting. We find that trust-region methods introduced in Sec. 2.3 naturally provide this proximal optimization by introducing the old policy $\pi_{\theta_{\text{old}}}$ to limit the magnitude of policy updates. We notice that there are two mainstream approaches. One approach is to apply a KL-Divergence penalty, which does not directly address the constraint on low-probability tokens. The clipping strategy, however, operates directly at the token level and explicitly provides less space for optimizing off-policy tokens, making it particularly simple yet effective at suppressing excessive updates to these tokens. Therefore, we adopt the clipping mechanism for its stability and stronger control over per-token update magnitudes. Specifically, the clipping strategy is implemented via a *policy ratio* for a given expert target $y^*$ and input $x$:

$$r(\theta) = \frac{\pi_\theta(y^* \mid x)}{\pi_{\theta_{\text{old}}}(y^* \mid x)}. \tag{5}$$

This ratio also quantifies how much the current policy has changed relative to the recent old policy for a specific action. To prevent aggressive updates, we directly moderate the supervised learning by clipping this ratio to a bounded interval $[1 - \epsilon, 1 + \epsilon]$, where the hyperparameter $\epsilon$ defines the extent of this trust region. Concurrently, recent work has also explored this strategy to avoid potential overfitting in SFT (Zhu et al., 2025). While reinforcement learning optimization objective uses a learned advantage estimate $\hat{A}$, SFT lacks an explicit reward signal. We bridge this gap by recognizing that the SFT objective implicitly treats every token in the dataset equivalent. Therefore, we set the advantage to a uniform positive constant, $\hat{A} = 1$, to uniformly encourage the adoption of all target behaviors. Thus, we get our Off-Policy token Clipped SFT (OPC-SFT) loss is:

$$\mathcal{L}_{\text{OPC-SFT}}(\theta) = -\mathbb{E}_{(x,y^*)\sim\mathcal{D}}\left[\min\left(r(\theta), \text{clip}(r(\theta), 1 - \epsilon, 1 + \epsilon)\right)\right]. \tag{6}$$

This loss ensures bounded policy updates. By adapting the already proven clipping strategy, it can constrain the SFT update's deviation from a periodically updated old policy, thereby stabilizing the training process. Specifically, if $r(\theta) > 1 + \epsilon$, indicating that the current policy overemphasizes $y^*$ in the expert dataset, clipping prevents an overly aggressive update which may lead to forgetting. Furthermore, periodically refreshing the old policy parameters $\theta_{\text{old}}$ allows this trust region to adapt as the model learns, balancing the acquisition of new, specialized knowledge with the retention of general capabilities. Consequently, OPC-SFT mitigates the destructive updates and preserves the model's prior knowledge, leading to more robust generalization.

## 4 EXPERIMENTS

We conduct a suite of experiments to demonstrate that OPC-SFT produces a robust cold-start policy that improves subsequent reinforcement learning performance compared to standard SFT and other strong baselines. Our evaluation primarily focuses on LLM agentic environments. First, in Sec. 4.1.1, we assess in-distribution generalization by testing the policy's ability to adapt to both seen and unseen task variations within the target domain. Second, to validate the anti-forgetting properties of OPC-SFT, we measure its out-of-distribution (OOD) performance on a set of general reasoning tasks, including code generation, mathematical problem-solving, and common-sense question-answering Sec. 4.1.2. Strong performance on these OOD tasks suggests the clipping mechanism effectively preserves the model's prior knowledge, a property that we believe contributes to its

Table 1: Performance on **ScienceWorld** and **ALFWorld** after cold-start. Metric is success rate(%). Best numbers are bolded.

| Backbone | Method | ScienceWorld | | ALFWorld | | Average |
| | | Seen | Unseen | Seen | Unseen | |
|---|---|---|---|---|---|---|
| **Qwen2.5-7B-Instruct** | ● SFT | 55.15 | 48.34 | 78.57 | 74.63 | 64.17 |
| | ● DFT | 57.73 | 51.18 | 75.71 | **79.85** | 66.12 |
| | ● NEFT | **58.76** | 50.24 | 73.57 | 74.63 | 64.30 |
| | ● OPC-SFT | 58.25 | **54.98** | 82.86 | 78.36 | **68.61** |
| **Qwen2.5-1.5B-Instruct** | ● SFT | 54.12 | 53.08 | 70.71 | 70.90 | 62.20 |
| | ● DFT | 64.43 | 56.40 | 61.43 | 70.90 | 63.29 |
| | ● NEFT | 60.82 | **58.77** | 62.14 | 69.40 | 62.78 |
| | ● OPC-SFT | **65.98** | 58.29 | **72.86** | **72.39** | **67.38** |
| **Llama3.2-3B-Instruct** | ● SFT | 56.70 | 53.55 | 75.00 | 70.90 | 64.04 |
| | ● DFT | **65.98** | 55.92 | 72.14 | 73.88 | 66.98 |
| | ● NEFT | 62.37 | 54.03 | **77.14** | 68.66 | 63.94 |
| | ● OPC-SFT | **65.98** | **64.93** | 76.43 | **77.61** | **71.24** |

superior performance after RL, as shown in Sec. 4.1.3. Third, to understand the mechanisms driving these performance gains, we conduct diagnostic analyses in Sec. 4.2 by visualizing the model's internal representations via PCA and tracking token probability progression. In Sec. 4.3, we investigate why OPC-SFT shows less pronounced gains on mathematical reasoning tasks. And we find the gradient norms of math data are smaller than those from the agentic tasks. Finally, in Sec. 4.4, we perform an ablation study on the clipping ratio $\epsilon$ to assess the robustness of OPC-SFT.

**Experimental Setup** Besides standard SFT, we also evaluate OPC-SFT against two baselines designed to improve SFT robustness. The first is a concurrent work DFT (Wu et al., 2025b) that rescales the SFT objective with the token probability. The second is NEFTune (Jain et al., 2023), a recent technique that improves performance by adding noise to embedding vectors during training. Our primary evaluation is conducted on the embodied agent environments of ALFWorld (Shridhar et al., 2021) and ScienceWorld (Wang et al., 2022). All models are trained and evaluated on a compute infrastructure equipped with accelerators capable of approximately 312 TFLOPS of BFloat16 (BF16) performance. And we select three models for evaluation, including Qwen2.5-7B-Instruct, Qwen2.5-1.5B-Instruct (Qwen et al., 2025) and Llama3.2-3B-Instruct (Grattafiori et al., 2024).

## 4.1 AGENTIC COLD START

Agentic tasks provide an ideal testbed for OPC-SFT. Succeeding in these environments requires the LLM to adopt a strict action format (Yao et al., 2024), which is often highly off-policy for a general-purpose model. Deviations from this format, such as generating semantically vague instructions like 'move somewhere' or syntactically invalid commands, can cause execution errors, terminate the environmental interaction, and lead to unpredictable agent behavior. Therefore, a robust SFT cold-start is essential for successfully initializing the LLM policy, teaching it the required format in a way that generalizes beyond the off-policy expert data.

## 4.1.1 IN-DISTRIBUTION VALIDATION BEFORE RL

We evaluate OPC-SFT on two agentic benchmarks: ALFWorld (Shridhar et al., 2021) and Science-World (Wang et al., 2022). A key advantage of these tasks is their setup for evaluating generalization within the target domain. They provide test sets with unseen instances that require the model to apply its learned knowledge to new scenarios that are variants of tasks during training. Specifically, the ALFWorld benchmark is composed of 140 seen and 134 unseen test samples, while Science-World contains 194 seen and 211 unseen samples. To ensure a fair and reproducible comparison, we adhere to standard evaluation protocols, using the framework from EMBod-Bench (Fei et al., 2025) for ALFWorld and ScienceWorld. As the result shown in Tab. 1, OPC-SFT achieves comparable performance against all the compelling methods.

Table 2: OOD performance under the **ALFWorld** setting. Methods: Base, SFT, DFT, NEFT, and **OPC-SFT**. Metrics are accuracy (%) and pass@1.

| Backbone | Method | MBPP | MMLU | HumanEval | GPQA | LiveCodeBench | MATH500 |
|---|---|---|---|---|---|---|---|
| Qwen2.5-7B-Instruct | Base | 79.68 | 71.00 | 73.03 | 33.84 | 61.54 | 76.80 |
| | SFT | 71.69 | 66.30 | 70.35 | 31.31 | 58.91 | 68.40 |
| | DFT | 75.13 | 70.20 | 74.88 | **34.85** | 60.02 | 69.00 |
| | NEFT | 72.22 | 70.10 | 43.90 | 28.79 | 63.01 | 69.60 |
| | **OPC-SFT** | **78.84** | **70.60** | **75.07** | 34.34 | **67.69** | **72.40** |
| Qwen2.5-1.5B-Instruct | Base | 58.73 | 60.08 | 69.63 | 21.72 | 14.50 | 52.60 |
| | SFT | 42.60 | 58.68 | 43.60 | 28.79 | 21.70 | 24.00 |
| | DFT | 45.50 | 58.63 | 44.40 | 9.60 | 24.52 | 19.20 |
| | NEFT | 46.31 | **59.17** | **45.01** | 28.52 | 27.04 | 18.80 |
| | **OPC-SFT** | **46.56** | 58.85 | 44.96 | **33.84** | **32.81** | **24.20** |
| Llama3.2-3B-Instruct | Base | 57.94 | 62.29 | 38.61 | 27.40 | 33.85 | 35.20 |
| | SFT | 56.61 | 58.47 | 44.39 | 10.61 | 34.50 | 21.20 |
| | DFT | 58.26 | 58.69 | 45.84 | 16.67 | 36.22 | 34.20 |
| | NEFT | 52.18 | 55.95 | 46.03 | 16.67 | 39.73 | 29.80 |
| | **OPC-SFT** | **58.71** | **59.97** | **48.13** | **18.18** | **42.60** | **37.60** |

Table 3: OOD performance under the **ScienceWorld** setting. Methods: Base, SFT, DFT, NEFT, and OPC-SFT. Metrics are accuracy (%) and pass@1.

| Backbone | Method | MBPP | MMLU | HumanEval | GPQA | LiveCodeBench | MATH500 |
|---|---|---|---|---|---|---|---|
| Qwen2.5-7B-Instruct | Base | 79.68 | 71.00 | 73.03 | 33.84 | 61.54 | 76.80 |
| | SFT | 67.20 | 64.43 | 64.63 | 46.46 | 49.55 | 56.40 |
| | DFT | 69.33 | 54.35 | 62.71 | 41.92 | 52.71 | 55.20 |
| | NEFT | 69.58 | **67.03** | 63.53 | 37.88 | 50.09 | 46.80 |
| | **OPC-SFT** | **71.96** | 66.64 | **69.13** | **63.64** | **55.59** | **57.40** |
| Qwen2.5-1.5B-Instruct | Base | 58.73 | 60.08 | 69.63 | 21.72 | 14.50 | 52.60 |
| | SFT | **49.74** | 58.31 | 41.28 | 30.81 | 7.73 | 42.80 |
| | DFT | 46.03 | 57.47 | 46.32 | **51.52** | 9.66 | 35.20 |
| | NEFT | 45.02 | 57.68 | 44.44 | 46.46 | 9.52 | 39.80 |
| | **OPC-SFT** | 48.15 | **58.62** | **47.82** | 50.51 | **11.83** | **43.40** |
| Llama3.2-3B-Instruct | Base | 57.94 | 62.29 | 38.61 | 27.78 | 33.85 | 35.20 |
| | SFT | 54.50 | 54.90 | 37.19 | 40.40 | 23.07 | 29.00 |
| | DFT | **58.11** | 55.95 | **39.12** | 50.00 | 24.59 | 32.80 |
| | NEFT | 55.03 | **57.19** | 32.31 | 45.96 | 25.15 | 31.20 |
| | **OPC-SFT** | 56.35 | 55.45 | 38.49 | **51.01** | **27.39** | **34.20** |

### 4.1.2 OUT-OF-DISTRIBUTION VALIDATION BEFORE RL

To evaluate knowledge retention and OOD generalization, we test the fine-tuned models on a suite of standard benchmarks. These include coding tasks, like MBPP (Austin et al., 2021) and HumanEval (Chen et al., 2021), general knowledge assessments, like MMLU (Hendrycks et al., 2021a) and GPQA (Rein et al., 2023), and mathematical reasoning MATH-500 (Hendrycks et al., 2021b). For GPQA, we use the GPQA Diamond subset. The results in Tab. 2 and Tab. 3 highlight the 'alignment tax' of standard SFT, which exhibits significant performance degradation. OPC-SFT, however, successfully preserves prior knowledge, cutting the performance degradation by an average of 11.54% relative to the SFT baseline. Notably, the Qwen2.5-7B-Instruct model trained with OPC-SFT consistently outperforms all other baselines, exhibiting the strongest anti-forgetting capabilities. This preservation of general knowledge is crucial for downstream agentic performance. Capabilities retained from pre-training, such as common-sense and logical reasoning, are beneficial for effective exploration and generalization within the agent's environment (Zhao et al., 2025). A high alignment tax actively limits the agent's potential to adapt to new situations, which is shown in Sec. 4.1.3. By reducing this tax, OPC-SFT provides a more capable and well-rounded foundation for the subsequent reinforcement learning phase.

### 4.1.3 PERFORMANCE COMPARISON AFTER RL

While a robust cold start is crucial, the ultimate measure of success is the performance of the RL-optimized model (DeepSeek-AI et al., 2025). We therefore take the policies initialized by each method and further train them using the established Group Relative Policy Optimization

Table 4: Final performance on **ScienceWorld** and **ALFWorld** after RL. Metric is success rate(%). Best numbers are bolded.

| Backbone | Method | ScienceWorld | | ALFWorld | | Average |
| | | Seen | Unseen | Seen | Unseen | |
|---|---|---|---|---|---|---|
| **Qwen2.5-7B-Instruct** | Base + GRPO | 41.75 | 47.87 | 62.86 | 58.96 | 52.86 |
| | SFT + GRPO | 60.82 | 60.19 | 85.00 | 76.87 | 69.97 |
| | DFT + GRPO | 61.34 | 59.24 | 90.71 | 80.06 | 72.84 |
| | NEFT + GRPO | 62.89 | 60.66 | 72.14 | 59.70 | 63.85 |
| | **OPC-SFT + GRPO** | **66.49** | **61.14** | **92.14** | **91.04** | **77.70** |
| **Qwen2.5-1.5B-Instruct** | Base + GRPO | 40.72 | 31.75 | 29.29 | 38.06 | 34.96 |
| | SFT + GRPO | 65.98 | 65.40 | 82.86 | 82.84 | 74.27 |
| | DFT + GRPO | **67.53** | 63.98 | 85.07 | 80.71 | 74.32 |
| | NEFT + GRPO | **67.53** | 63.51 | 82.86 | 67.16 | 70.27 |
| | **OPC-SFT + GRPO** | 65.46 | **68.72** | **90.00** | **94.03** | **79.55** |
| **Llama3.2-3B-Instruct** | Base + GRPO | 44.85 | 44.08 | 0.00 | 0.00 | 22.23 |
| | SFT + GRPO | 70.62 | 63.03 | 92.86 | 89.55 | 79.02 |
| | DFT + GRPO | 67.53 | 64.45 | 81.43 | 76.87 | 72.57 |
| | NEFT + GRPO | 61.86 | 61.86 | 67.86 | 45.52 | 59.28 |
| | **OPC-SFT + GRPO** | **73.71** | **65.40** | **94.29** | **92.54** | **81.49** |

(GRPO) algorithm (Shao et al., 2024). The final LLM-based agent performance, detailed in Tab. 4, demonstrates the significant downstream benefits of OPC-SFT. The LLM initialized with OPC-SFT achieves a substantial performance advantage over all baselines, particularly on the ALFWorld benchmark. The base model struggles when fine-tuned directly with GRPO, demonstrating that cold-start is necessary in agentic tasks.

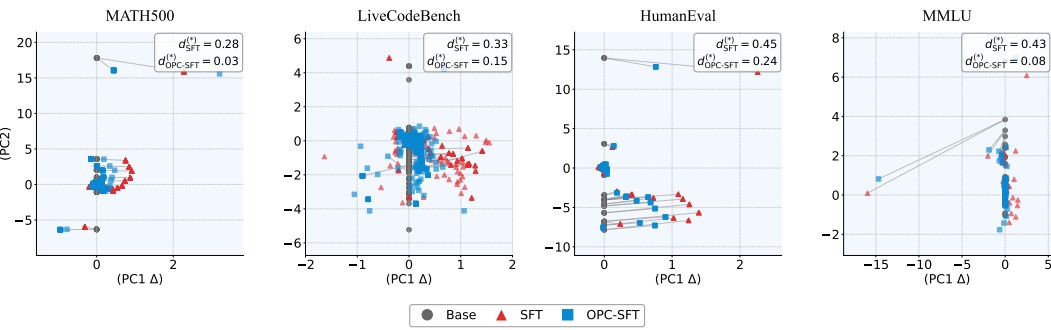

Figure 3: PCA shift of Llama3.2-3B-Instruct with the SFT and OPC-SFT methods.

## 4.2 Latent Space shift analysis and Target Token Probability change over Training Steps

We conduct internal representation and gradient analysis to account for the advantages of OPC-SFT. Xu et al. (2025) have shown that PCA shift analysis serves as a sensitive and interpretable metric for representational changes associated with task performance. We decompose the latent space of the LLM on the target domain using PCA ($n = 2$) projection. We can clearly observe from Fig. 3 that OPC-SFT exhibits a smaller divergence from the base model compared to SFT.

Specifically, when evaluated by Euclidean distance $d^{(*)} = \|\mathbf{z}^{(*)} - \mathbf{z}^{(\mathrm{orig})}\|_2$, where $\mathbf{z}$ denotes the mean PCA coordinates of hidden states across layers in the low-dimensional space. SFT yields divergences of 0.28, 0.33, 0.45, and 0.43 on the MATH500, LiveCodeBench, HumanEval, and MMLU benchmarks, respectively. In contrast, OPC-SFT significantly reduces these divergences, yielding 0.03, 0.15, 0.24, and 0.08 on the same respective benchmarks. The projection details are deferred to Appx. D.1. Furthermore, analysis of the target token probability progression, as seen in Fig. 4, reveals that OPC-SFT increases target token probabilities more steadily.

## 4.3 Other tasks like Math

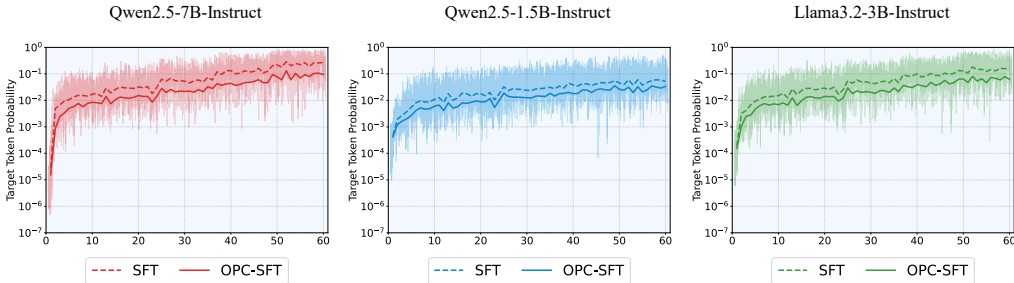

Figure 4: Token probability in the bottom 1% quantile over training steps for SFT and OPC-SFT.

Table 5: Final mathematical reasoning performance after the RL phase. LLMs initialized with different cold-start methods are trained with GRPO. Metrics are accuracy (%) and avg@8.

| Backbone | Method | Minerva | Olympiad Bench | GSM8K | AIME24 | MATH500 | Average |
|----------|--------|---------|----------------|-------|--------|---------|---------|
| Qwen2.5-7B-Instruct | SFT + GRPO | **39.71** | 47.45 | 94.31 | 21.67 | 84.00 | 57.43 |
| | OPC-SFT + GRPO | 39.34 | **48.33** | **95.68** | **23.33** | **84.35** | **58.21** |
| Qwen2.5-1.5B-Instruct | SFT + GRPO | **30.52** | 38.56 | 84.95 | **17.50** | **77.30** | 49.77 |
| | OPC-SFT + GRPO | 29.81 | **40.74** | **85.97** | 15.41 | 77.20 | **49.83** |
| Llama3.2-3B-Instruct | SFT + GRPO | 19.71 | 22.08 | 79.88 | **9.58** | 56.60 | 37.57 |
| | OPC-SFT + GRPO | **20.59** | **22.51** | **80.54** | 9.55 | **57.10** | **37.86** |

To investigate mathematical reasoning performance, we fine-tune the LLM on data collected from the DeepScaleR (Luo et al., 2025b) problem suite and report the final performance in Tab. 5. While OPC-SFT still outperforms the baseline, the performance gains are more modest than those in the agentic tasks. This finding, which is consistent with concurrent work (Zhu et al., 2025), prompts us to investigate the conditions under which OPC-SFT is most effective. We hypothesize that the performance discrepancy is due to the nature of the fine-tuning data. Unlike the novel action formats in agentic tasks, mathematical reasoning is already well-represented in the LLMs' pre-training corpora. Consequently, SFT for math involves a smaller distributional shift, resulting in less of the off-policy instability that

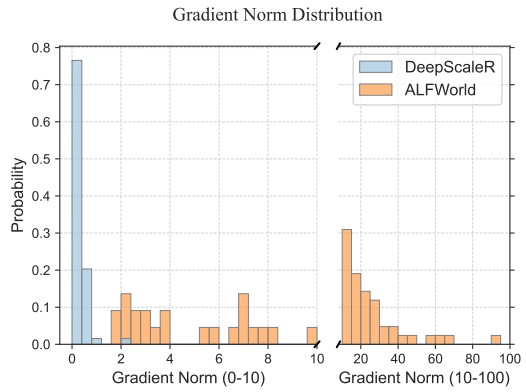

Figure 5: Gradient Norm Comparison between math task DeepScaleR and agentic task ALFWorld.

OPC-SFT is designed for. To test this hypothesis, we analyze the gradient norm distributions at the beginning of the SFT phase for both task types. The results show that the gradient norms for the agentic task data are substantially larger than those for the math reasoning data. This evidence demonstrates that the benefits of OPC-SFT are pronounced when the fine-tuning data is off-policy.

### 4.4 ABLATION STUDY ON DIFFERENT CLIP RATIO

We conduct an ablation study to understand the influence of the clipping ratio $\epsilon$ on final task performance. Using the Llama3.2-3B-Instruct models, we vary $\epsilon$ and measure the impact on performance. As shown in Fig. 6, this method mainly outperforms standard SFT within the range of $[0.4, 0.6]$, although a slight decrease in performance is observed at the boundaries of this range. This demonstrates the robustness of OPC-SFT, as even a suboptimal choice of the clip ratio within this effective range yields gains over the baseline and does not lead to performance degradation.

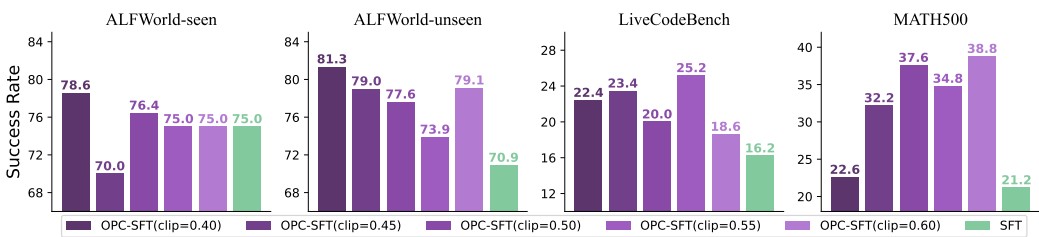

Figure 6: Effect of the clipping ratio on Llama3.2-3B-Instruct's performance on ALFWorld.

Table 6: OOD performance for the old policy update frequency ablation.

| $\pi_{\theta_{\text{old}}}$ Update Freq | MBPP | MMLU | HumanEval | GPQA | LiveCodeBench | MATH500 |
|---|---|---|---|---|---|---|
| 128 | 58.99 | 57.84 | 47.98 | 17.17 | 36.65 | 35.60 |
| 256 | 58.71 | **59.97** | 48.13 | 18.18 | **42.60** | **37.60** |
| 512 | **61.11** | 57.03 | **48.89** | 19.70 | 36.79 | 36.20 |
| 1024 | 60.05 | 59.26 | 46.53 | **21.21** | 35.68 | 36.40 |

## 4.5 ABLATION STUDY ON OLD POLICY UPDATE FREQUENCY

A critical component of our optimization framework is the old policy model $\pi_{\theta_{\text{old}}}$. As training progresses, the current policy $\pi_\theta$ naturally diverges from earlier states. Hence, we conduct an ablation study on the update frequency, defined as the number of samples used to train the policy model before resetting the old policy. We evaluate Llama3.2-3B-Instruct using intervals of 128, 256, 512, and 1024 samples. As shown in Tab. 6 and Fig. 7, larger intervals lead to substantial degradation,

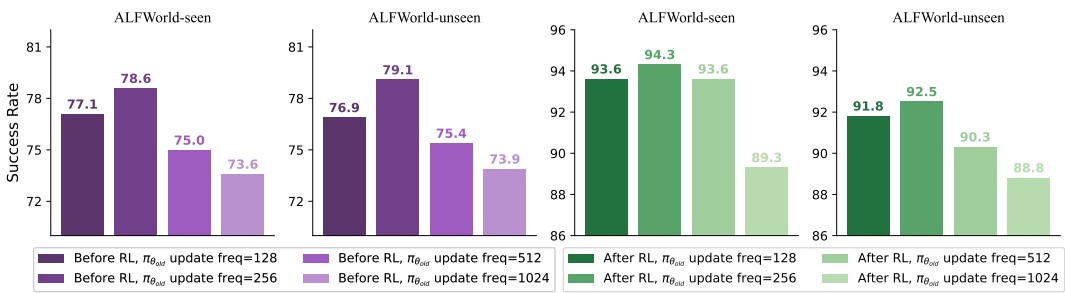

Figure 7: Old policy update frequency ablation on Llama3.2-3B-Instruct for the ALFWorld Task

confirming that a 'stale' old policy over-constrains the optimization and prevents the model from adopting improved behaviors. Conversely, updating too frequently yields slightly worse performance, likely because the trust region shifts too rapidly to provide a stable old policy.

## 5 CONCLUSIONS

SFT cold-start yields an initial policy for reinforcement learning in LLMs. For efficient subsequent RL, the initial model requires a delicate balance: learning new, specialized skills alongside the robust retention of vast prior knowledge. In this work, we addressed a key challenge during SFT cold-start: the degradation of generalization and pre-trained knowledge. We find this problem to be particularly acute when fine-tuning on specialized, off-policy datasets that are totally novel compared with the model's pre-training corpus. Additionally, we identify the cause of this forgetting mainly as the large gradient norm in the initial stage brought by off-policy tokens. Hence, we propose OPC-SFT for a robust cold-start, which clips the update of off-policy tokens. We have demonstrated OPC-SFT's strong generalization and anti-forgetting ability, which prepares LLMs for RL. While OPC-SFT demonstrates clear benefits, we acknowledge its limitation. If the clipping is set too strictly, it could over-constrain updates from off-policy tokens, potentially leading to overly aggressive updates on medium-probability tokens. This, in turn, might lead to a collapse in model entropy.

## REPRODUCIBILITY STATEMENT

In this study, to ensure the reproducibility of this paper, we provide key information from our submission as follows.

1. **Training Algorithm.** We provide our approach in Sec. 3.3.
2. **Source Code and Data.** We have submitted the source code of OPC-SFT in the supplementary materials. ALFWorld training data is available at https://huggingface.co/LEVI-Project/sft-data. For ScienceWorld, we use the data in https://huggingface.co/datasets/AgentGym/AgentTraj-L/tree/main.
3. **Experimental Details.** We list the detailed experiment settings, computational resources. And hyperparameters in Appx. D.4.
4. **Derivation Details.** We provide the missing proofs in Appx. A.

## ETHICS STATEMENT

The authors confirm their adherence to the Code of Ethics. This research is purely methodological and does not involve human subjects or applications with foreseeable negative societal impacts. We are committed to keeping transparency and integrity throughout the research process.

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

## THE USE OF LLMS

In the preparation of this manuscript, we employed large language models (LLMs) as a general-purpose writing aid. Specifically, their use was confined to minor language polishing, including grammar correction and improvement of sentence structure, to enhance the overall readability of the text. The LLMs did not contribute to any of the core research aspects of this work, such as the formulation of ideas, the design of algorithms, theoretical derivations, or the execution and analysis of experiments. The intellectual content and all claims made within this paper are solely the work of the human authors, who bear full responsibility for the final manuscript.

## A    CLIPPING STRATEGY DETAILS

We present the analysis of the clipping mechanisms in OPC-SFT. To ground this analysis, we define the optimization context on a supervised learning dataset $\mathcal{D} = \{(x, y^*)\}$. Drawing inspiration from the well-established clipping strategy in PPO (Schulman et al., 2017), we implement a token-level trust region to explicitly manage off-policy tokens and prevent destructive policy updates.

The gradient of the OPC-SFT loss inherits the stabilizing behavior of the trust region method:

$$\nabla_\theta \mathcal{L}_{\text{OPC-SFT}}(\theta) = -\mathbb{E}_{(x,y^*)\sim\mathcal{D}} \left[\nabla_\theta \left(\min\left(r(\theta),\ \text{clip}(r(\theta), 1-\epsilon, 1+\epsilon)\right)\right)\right].$$

Here, $r(\theta)$ represents the probability ratio for a specific token. This gradient behaves in two distinct cases:

When $r(\theta) \leq 1 + \epsilon$, the gradient uses $r(\theta)$, enabling meaningful learning:

$$\nabla_\theta r(\theta) = r(\theta) \cdot \nabla_\theta \log \pi_\theta(y^* \mid x).$$

When $r(\theta) > 1 + \epsilon$, the clipped term dominates the $\min$, so the gradient is zero:

$$\nabla_\theta \left(\text{clip}(r(\theta), 1-\epsilon, 1+\epsilon)\right) = 0.$$

By strictly enforcing this token-level constraint, OPC-SFT suppresses gradients for samples where the policy deviates too far from the old policy, preventing the aggressive updates often associated with off-policy tokens.

## B    ADDITIONAL EXPERIMENTS

### B.1    OPC-SFT ON MATH REASONING TASKS

We show the performance of OPC-SFT and SFT on mathematical reasoning benchmarks, as shown in Tab. 7 and their generalization to common-sense question-answering MMLU (Hendrycks et al., 2021a), GPQA (Rein et al., 2023), and coding tasks HumanEval (Chen et al., 2021) and Live-CodeBench (Jain et al., 2025), as shown in Tab. 8. This shows that the benefits of OPC-SFT are most pronounced when the expert data is highly off-policy, which could result in large gradient norms at the initial stage of cold-start.

Table 7: Mathematical reasoning performance of the cold-start policies (before RL). OPC-SFT is compared against the standard SFT baseline. Metrics are accuracy (%) and avg@8.

| Backbone | Method | Minerva | Olympiad Bench | GSM8K | AIME24 | MATH500 | Average |
|---|---|---|---|---|---|---|---|
| Qwen2.5-7B-Instruct | Base | 38.24 | 36.37 | 91.89 | 13.75 | 76.80 | 51.41 |
| | SFT | 39.04 | 42.37 | **94.24** | **18.33** | **81.80** | 55.16 |
| | **OPC-SFT** | **41.48** | **43.56** | 93.78 | 15.83 | 81.40 | **55.21** |
| Qwen2.5-1.5B-Instruct | Base | 13.76 | 18.07 | 70.58 | 1.67 | 52.60 | 31.34 |
| | SFT | 24.26 | 36.89 | 82.11 | **10.86** | 74.40 | 45.70 |
| | **OPC-SFT** | **26.16** | **38.11** | **84.91** | 10.00 | **76.60** | **47.16** |
| Llama3.2-3B-Instruct | Base | 10.56 | 9.63 | 67.25 | 0.83 | 35.20 | 24.69 |
| | SFT | 14.31 | 18.07 | 76.63 | 1.67 | **52.20** | 32.58 |
| | **OPC-SFT** | **16.54** | **18.34** | **77.15** | **3.34** | 51.80 | **33.43** |

Table 8: OOD performance of models for different cold-start. OPC-SFT + GRPO is compared against SFT + GRPO baseline. Metrics are accuracy (%) and pass@1.

| Backbone | Method | MMLU | HumanEval | GPQA | LiveCodeBench | Average |
|---|---|---|---|---|---|---|
| **Qwen2.5-7B-Instruct** | Base | 71.00 | 73.03 | 33.84 | 61.54 | 59.85 |
| | SFT | 64.43 | 1.07 | 35.35 | 0.20 | 25.26 |
| | **OPC-SFT** | **66.64** | **2.74** | **35.35** | **0.31** | **26.26** |
| **Qwen2.5-1.5B-Instruct** | Base | 60.08 | 69.63 | 21.72 | 14.50 | 41.48 |
| | SFT | 58.31 | 0.00 | 27.40 | 0.00 | **21.43** |
| | **OPC-SFT** | **58.62** | 0.00 | 26.77 | 0.00 | 21.35 |
| **Llama3.2-3B-Instruct** | Base | 62.29 | 38.61 | 27.78 | 33.85 | 40.63 |
| | SFT | 54.90 | 16.84 | 30.30 | 0.00 | 25.51 |
| | **OPC-SFT** | **55.45** | **17.37** | **31.31** | 0.00 | **26.03** |

Table 9: SFT and SFT+RL Performance on **ScienceWorld** and **ALFWorld**. Metrics are success rate (%). Best numbers are bolded.

| Backbone | Method | ScienceWorld | | ALFWorld | |
|---|---|---|---|---|---|
| | | Before RL | After RL | Before RL | After RL |
| **Qwen2.5-7B-Instruct** | SFT | 34.44 | 43.33 | 76.87 | 84.33 |
| | DFT | 37.78 | 44.44 | 75.37 | 80.60 |
| | NEFT | 41.11 | 48.89 | 77.61 | 64.92 |
| | OPC-SFT | **50.00** | **58.89** | **79.10** | **90.29** |
| **Qwen2.5-1.5B-Instruct** | SFT | 32.22 | 45.56 | 73.88 | 82.08 |
| | DFT | 30.00 | 50.00 | 67.91 | 82.83 |
| | NEFT | 37.78 | 53.33 | 73.13 | 82.83 |
| | OPC-SFT | **40.00** | **55.56** | **76.12** | **87.31** |
| **Llama3.2-3B-Instruct** | SFT | 37.78 | 41.11 | 72.39 | 83.58 |
| | DFT | 41.11 | 46.67 | 72.39 | 55.00 |
| | NEFT | 43.33 | 44.44 | 70.90 | 56.71 |
| | OPC-SFT | **53.33** | **61.10** | **76.87** | **91.79** |

## B.2 AGENTIC TASK WITH AGENTBOARD VALIDATION DATASET

Besides EMBod-Bench (Fei et al., 2025) used in Sec. 4.1.1, we also adopt the AgentBoard (Ma et al., 2024) framework to conduct a two-stage experiment: first performing SFT, followed by RL, aiming to investigate the cold-start performance of different SFT approaches. AgentBoard is a benchmark for evaluating multi-turn LLM agents. It spans nine task categories and over a thousand environments, encompassing widely used benchmarks such as ALFWorld and ScienceWorld, which capture multi-round and partially observable settings. Its accompanying open-source toolkit further enables detailed analysis through visualization of trajectories, skill-specific performance, and difficulty breakdowns, providing a comprehensive diagnostic framework for agent research.

We conduct experiments on the ALFWorld and ScienceWorld benchmarks. Compared to EMBod-Bench (Fei et al., 2025), the test sets of these two benchmarks in the AgentBoard framework differ as follows: ALFWorld contains 134 unseen test instances that overlap with those in EMBod-Bench, whereas ScienceWorld includes 90 unseen test instances that are distinct from those in EMBod-Bench. In addition, the inference settings of AgentBoard and EMBod-Bench are not identical. The results are shown in Tab. 9.

ALFWorld (Shridhar et al., 2021) consists of planning tasks situated in household settings, ranging from basic object manipulation (e.g., pick-and-place) to scenarios that demand multi-step interactions. For instance, in the "discard a card" task, the agent must first identify the target card, pick it up, locate a trash bin, and correctly dispose of the card to complete the task.

ScienceWorld (Wang et al., 2022) is a challenging benchmark that requires models to carry out scientific experiments in a interactive environment. The environment is supported by a physics engine that incorporates thermodynamic and electrical systems, thus demanding strong planning and causal reasoning skills. For example, one task may ask: turn on the red light bulb by powering it using a renewable power source.

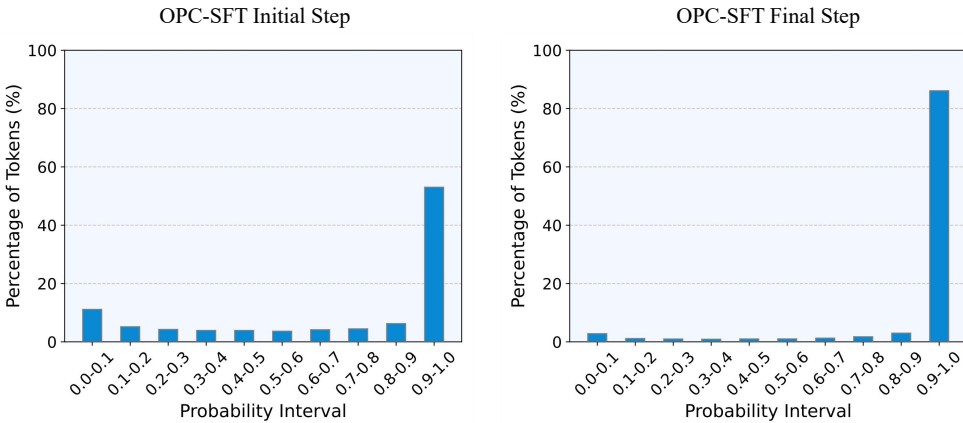

Figure 8: Target token probability distribution at the beginning and end of OPC-SFT. The x-axis shows probability intervals, and the y-axis shows the proportion of tokens in each interval.

### B.3 TARGET TOKEN DISTRIBUTION DURING OPC-SFT

Supervised Fine-Tuning is widely used to enhance the performance of Large Language Models on task-specific objectives. In SFT, the model is trained on a dataset of high-quality input-output pairs, which are typically derived from expert demonstrations or synthetic trajectories generated by teacher models. Through this process, the LLM learns structured reasoning patterns, task-specific knowledge, and preferred action strategies.

To illustrate the effect of OPC-SFT, we analyze the distribution of token-level probabilities at the beginning and at the end of training. We divide the probability, which range from 0 to 1, into discrete intervals and plot the proportion of target tokens falling into each interval. In the initial stage of training, a noticeable fraction of target tokens have low probability, indicating uncertain predictions. By the end of training, the distribution shifts significantly: nearly all target tokens attain higher probability, reflecting increased confidence and better alignment with the expert trajectories as shown in Fig. 8. This visualization quantitatively demonstrates how SFT improves the model's certainty and task-specific performance.

### B.4 TRAINING REWARD CURVE IN REINFORCEMENT LEARNING

Large Language Models first acquire general reasoning and task-specific patterns through SFT, providing a strong and high-performing initialization for subsequent RL. In this study, after SFT, we further train LLMs in two benchmark environments: ALFWorld and ScienceWorld, to adapt the pretrained models to interactive, sequential decision-making tasks. This two-stage training paradigm allows the models to start from a higher baseline, which facilitates more effective exploration and accelerates policy refinement through trial-and-error interactions in the environment. Each LLM is trained using one of four strategies: SFT, DFT, NEFT, and OPC-SFT, where OPC-SFT incorporates on-policy correction during RL to improve stability and sample efficiency.

Fig. 9 shows the normalized training rewards over the first 100 steps. The top row corresponds to ALFWorld, and the bottom row corresponds to ScienceWorld. Each subplot contains four curves representing the different training strategies: SFT, DFT, NEFT, and OPC-SFT. Here, the reward indicates the success of a trajectory: 1 for success and 0 for failure. Solid lines represent smoothed rewards, while semi-transparent lines show raw values.

From Fig. 9, it is clear that OPC-SFT consistently achieves higher rewards across both environments. Starting from a strong SFT initialization gives OPC-SFT a higher starting point, which not only accelerates early performance but also encourages more effective exploration, enabling the model to discover successful trajectories faster. These results highlight the importance of combining supervised pre-training with on-policy RL correction: the LLMs first acquire structured reasoning and task knowledge via SFT, and then efficiently adapt their policies to maximize task success through

RL. Overall, this two-stage approach enables LLMs to leverage prior knowledge while learning interactive behaviors, achieving both sample-efficient learning and robust task performance.

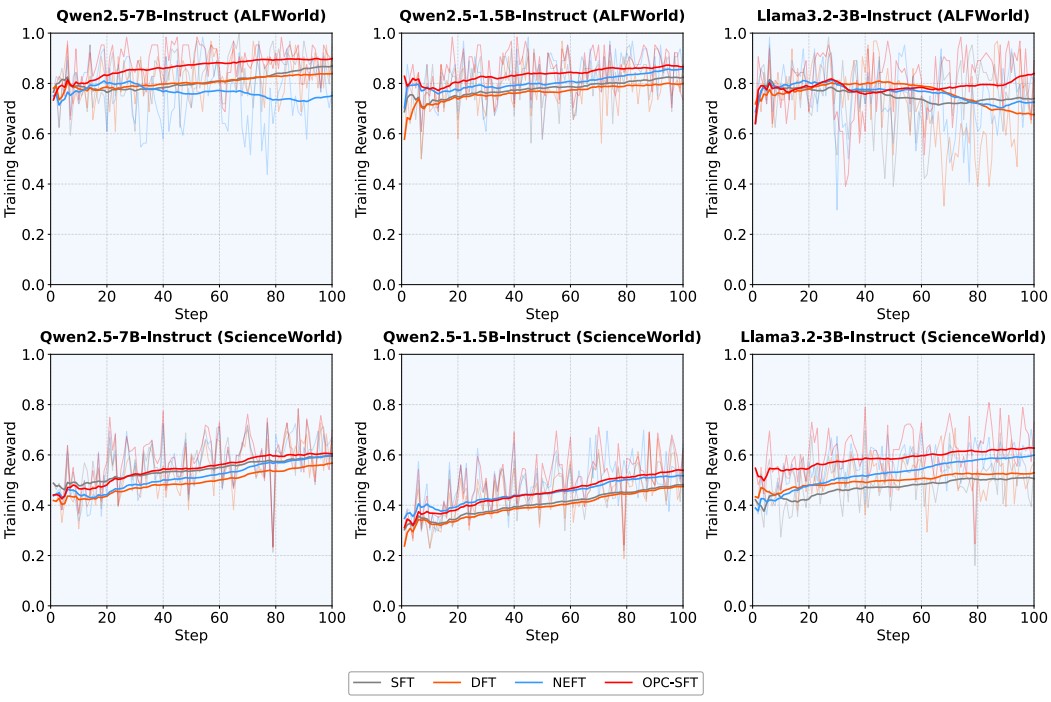

Figure 9: Training reward comparison across models and environments. The top row shows results in the ALFWorld environment for three models, and the bottom row shows results in the Science-World environment. Each subplot contains four curves corresponding to SFT, DFT, NEFT, and OPC-SFT. Solid lines represent smoothed reward values calculated as a running average, and semi-transparent lines show the raw rewards recorded during training. The x-axis denotes training steps, and the y-axis denotes the normalized reward.

### B.5 The Changes in Clipped Token Counts during Old Policy Updates

In OPC-SFT, the old policy $\pi_{\theta_{old}}$ is periodically updated, and the percentage of clipped tokens decreases as training progresses. Eventually, the policy will converge, and $\frac{\pi_\theta}{\pi_{\theta_{old}}}$ will approximate 1, not exceeding $1 + \epsilon$. As a result, the number of clipped tokens decreases as the gradient magnitude gradually becomes smaller until convergence. We provide the corresponding figure for ALFWorld with $\epsilon = 0.5$, as shown in Fig. 10. If the clipping ratio is set to a larger value, the number of clipped tokens will decrease further.

## C LLM Fine-Tuning Related work

LLMs have demonstrated a strong capacity for multi-step reasoning, a crucial component for solving complex problems (Zhao et al., 2023). This capability is rooted in their pre-training on extensive and diverse corpora (Qwen et al., 2025; Touvron et al., 2023). While high-quality pre-training data is critical for shaping these foundational abilities, it is often insufficient for specialized, challenging domains. Agentic tasks, for example, demand complex reasoning that is deeply integrated with planning and executing actions in an interactive environment (Wu et al., 2025a). Therefore, post-training is essential to adapt LLMs to these specific domains, significantly enhancing their ability to perform such intricate tasks (Wang et al., 2025; Team et al., 2025).

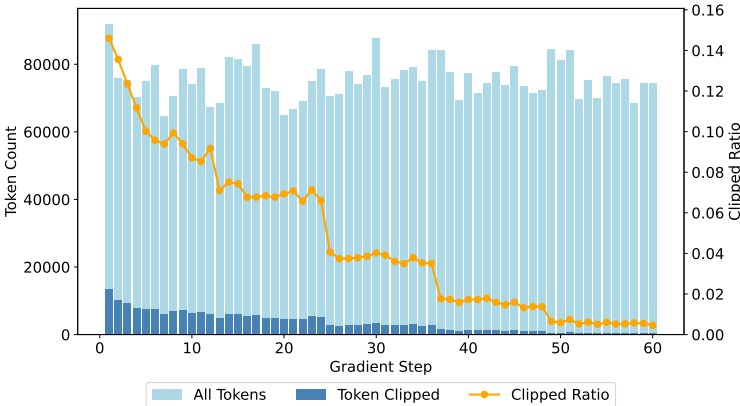

Figure 10: Changes in clipped tokens over steps when using OPC-SFT with $\epsilon = 0.5$ on ALFWorld

## C.1 SUPERVISED FINE-TUNING

SFT is a foundational post-training stage that significantly enhances LLMs by aligning them with human instructions. By training on high-quality prompt-response pairs, SFT refines the model's ability to generate coherent and contextually appropriate outputs. However, this process introduces a critical trade-off: while extensive SFT improves instruction-following, it can also reduce the diversity of the model's generations (Li et al., 2025; Wang et al., 2024). Over-optimization on a fixed set of responses may lead to mode collapse, where the model consistently produces similar outputs, thereby limiting its exploratory capabilities. This loss of diversity is particularly detrimental for downstream reinforcement learning, where a broad search space is essential for discovering optimal policies (Zeng et al., 2025). Striking a balance between alignment and diversity is thus a key challenge, as excessive fine-tuning risks narrowing the model's generative flexibility. Indeed, recent studies suggest that SFT can substantially alter the LLM's latent space, limiting transferability (Huan et al., 2025). Consequently, some approaches bypass SFT entirely, using direct reinforcement learning to enhance exploration and improve reasoning (DeepSeek-AI et al., 2025; Zeng et al., 2025). There are also concurrent works seeking methods to enhance generalization for SFT (Zhu et al., 2025; Wu et al., 2025b). While our method shares the operational concept of integrating trust regions into SFT with concurrent work (Zhu et al., 2025), our motivation differs significantly. Our approach is driven by the insight that off-policy tokens, characterized by large gradient norms, are the primary drivers of catastrophic forgetting.

## C.2 REINFORCEMENT LEARNING

Building on the framework of Reinforcement Learning from Human Feedback (RLHF), recent studies have extended RL to enhance the reasoning capabilities of LLMs (Trung et al., 2024; Kazemnejad et al., 2024; Gehring et al., 2024). Beyond its application in mathematical reasoning, RL provides a general mechanism for optimizing non-differentiable objectives, aligning models with human preferences, and encouraging effective exploration of solution spaces. By directly shaping model behavior through reward signals, RL complements supervised training and enables LLMs to achieve improved generalization and robustness. Nevertheless, applying standard algorithms such as PPO is resource-intensive, as it requires an additional critic network, substantially increasing computational cost and GPU memory usage. To alleviate this, ReMax (Li et al., 2024) employs the REINFORCE algorithm with greedy sampling as a reward baseline. Group Relative Policy Optimization (GRPO) (Shao et al., 2024) introduces a more memory-efficient variant of PPO that enhances reasoning performance. Reinforce++ (Hu, 2025) integrates techniques such as PPO clipping and reward normalization to improve stability and training efficiency. Furthermore, since policy entropy tends to diminish rapidly during training, reducing exploration, DAPO (Yu et al., 2025b) proposes the Clip-Higher strategy to counteract this effect.

### C.3 CONTINUAL LEARNING

Continual Learning (CL) aims to enable models to learn from a stream of tasks sequentially without suffering from catastrophic forgetting (Lopez-Paz & Ranzato, 2017; Kirkpatrick et al., 2017; De Lange et al., 2022). In the context of LLM fine-tuning, CL research primarily focuses on multi-task learning, investigating how to adapt models to new domains or instructions without erasing learned specific abilities for previous tasks (Wu et al., 2024). In current mainstream approaches, replay-based methods alleviate forgetting by storing a small subset of data from previous tasks in a memory buffer and revisiting them during training to maintain historical knowledge (Scialom et al., 2022). Regularization-based approaches constrain parameter updates to prevent interference with knowledge acquired from previous tasks (Razdaibiedina et al., 2023). While continual learning typically targets the maintenance of performance across a sequence of distinct supervised tasks, our work primarily focuses on retaining general pre-trained capabilities, most notably reasoning and prior knowledge, during the SFT cold-start phase.

## D EXPERIMENTAL DETAILS

### D.1 PCA DETAILS

Given a batch of queries $x$, we extract hidden states $H_i^{(*)}(x)$ at each layer $i$ for both model states $(*) \in \{\text{orig}, \text{upd}\}$. Principal Component Analysis (PCA) with $n = 2$ is then applied to $H_i^{(*)}$, and the mean projections onto the first and second principal directions (PC1 and PC2) are denoted by $m_{i,1}^{(*)}$ and $m_{i,2}^{(*)}$, respectively. The shift along PC1 is defined as

$$\Delta m_{i,1}^{(*)} = m_{i,1}^{(*)} - m_{i,1}^{(\text{orig})},$$

while $m_{i,2}^{(*)}$ is reported for PC2 as an auxiliary indicator of distributional variation, with smaller values reflecting more stable representations. For each model state $(*)$, we define a representation center as the mean of PCA coordinates across all layers:

$$\mathbf{z}^{(*)} = \frac{1}{L} \sum_{i=1}^{L} \mathbf{z}_i^{(*)}, \quad \mathbf{z}_i^{(*)} = \left( \Delta m_{i,1}^{(*)}, m_{i,2}^{(*)} \right),$$

where $L$ denotes the total number of layers. Finally, the latent shift between state $(*)$ and the original (base) model is quantified by the Euclidean distance

$$d^{(*)} = \|\mathbf{z}^{(*)} - \mathbf{z}^{(\text{orig})}\|_2.$$

### D.2 ALFWORLD DEMONSTRATION

To concretely illustrate the reasoning capabilities of LLMs, we present an example interaction within the ALFWorld environment. In this demonstration, each prompt is constructed in a multi-turn manner, combining the previous response with the current observation to inform the LLM's subsequent reasoning and actions. The dialogue below highlights the step-by-step interaction process, showing how the LLM produces structured outputs such as thoughts and actions, and reflecting its iterative reasoning dynamics in an embodied setting.

> **System**
>
> Your are an expert in the ALFRED Embodied Environment.

**User**

Interact with a household to solve a task. Imagine you are an intelligent agent in a household environment and your target is to perform actions to complete the task goal. At the beginning of your interactions, you will be given the detailed description of the current environment and your goal to accomplish.
For each of your turn, you will be given the observation of the last turn. You should first think about the current condition and plan for your future actions, and then output your action in this turn. Your output must strictly follow this format:"Thought: your thoughts.\nAction: your next action".

The available actions are:
1. go to {recep}
2. task {obj} from {recep}
3. put {obj} in/on {recep}
4. open {recep}
5. close {recep}
6. toggle {obj} {recep}
7. clean {obj} with {recep}
8. heat {obj} with {recep}
9. cool {obj} with {recep}
where {obj} and {recep} correspond to objects and receptacles.
After your each turn, the environment will give you immediate feedback based on which you plan your next few steps. if the envrionment output "Nothing happened", that means the previous action is invalid and you should try more options.

Your response should use the following format:

Thought: <your thoughts>
Action: <your next action>

**Assistant**

OK

**User**

You are in the middle of a room. Looking quickly around you, you see a bathtubbasin 1, a cabinet 2, a cabinet 1, a countertop 1, a garbagecan 1, a handtowelholder 1, a sinkbasin 1, a toilet 1, a toiletpaperhanger 1, and a towelholder 1. Your task is to: put a toiletpaper in toiletpaperhanger.

**Assistant**

Thought: I need to place a toilet paper roll on the toilet paper hanger. First, I should check if there's already a roll on the hanger or if it's empty.
Action: go to toiletpaperhanger 1

**User**

Observation: On the toiletpaperhanger 1, you see nothing.

> **Assistant**
>
> Thought: The toilet paper hanger is empty, so I need to find a new roll of toilet paper to put on it.
> Action: go to toilet 1

> **User**
>
> Observation: On the toilet 1, you see a soapbottle 1, and a toiletpaper 1.

> **Assistant**
>
> Thought: I've found a toilet paper roll on the toilet. I should pick it up so I can place it on the hanger.
> Action: take toiletpaper 1 from toilet 1

> **User**
>
> Observation: You pick up the toiletpaper 1 from the toilet 1.

> **Assistant**
>
> Thought: Now that I have the toilet paper roll, I need to return to the toilet paper hanger to install it.
> Action: go to toiletpaperhanger 1

> **User**
>
> Observation: On the toiletpaperhanger 1, you see nothing.

> **Assistant**
>
> Thought: I'm back at the toilet paper hanger with the roll in hand. It's time to put the toilet paper on the hanger.
> Action: put toiletpaper 1 in/on toiletpaperhanger 1

### D.3 SCIENCEWORLD DEMONSTRATION

To further illustrate the reasoning capabilities of LLMs across environments, we provide the initial prompt for a task in the ScienceWorld environment. For brevity, only the starting prompt is shown here; the subsequent interaction process is analogous to the ALFWorld demonstration, which is described in Sec. D.3, and is omitted.

> **System**
>
> You are a helpful agent that interacts with the virtual science school environment to solve the given task.

**User**

You are an agent for ScienceWorld. Every round I will give you an observation, and you have to respond with an action based on the observation to finish the given task. Here are the actions you may take:["action": "open/close OBJ", "description": "open/close a container", "action": "de/activate OBJ", "description": "activate/deactivate a device", "action": "connect OBJ to OBJ", "description": "connect electrical components", "action": "disconnect OBJ", "description": "disconnect electrical components", "action": "use OBJ [on OBJ]", "description": "use a device/item", "action": "look around", "description": "describe the current room", "action": "look at OBJ", "description": "describe an object in detail", "action": "look in OBJ", "description": "describe a container's contents", "action": "read OBJ", "description": "read a note or book", "action": "move OBJ to OBJ", "description": "move an object to a container", "action": "pick up OBJ", "description": "move an object to the inventory", "action": "put down OBJ", "description": "drop an inventory item", "action": "pour OBJ into OBJ", "description": "pour a liquid into a container", "action": "dunk OBJ into OBJ", "description": "dunk a container into a liquid", "action": "mix OBJ", "description": "chemically mix a container", "action": "go to LOC", "description": "move to a new location", "action": "eat OBJ", "description": "eat a food", "action": "flush OBJ", "description": "flush a toilet", "action": "focus on OBJ", "description": "signal intent on a task object", "action": "wait", "description": "take no action for 10 iterations", "action": "wait1", "description": "take no action for 1 iteration", "action": "task", "description": "describe current task", "action": "inventory", "description": "list your inventory"]
Your response should use the following format:
Thought:
your thoughts.

Action:
your next action

**Assistant**

OK. I'll follow your instructions and try my best to solve the task.

**User**

Your task is to boil water. For compounds without a boiling point, combusting the substance is also acceptable. First, focus on the substance. Then, take actions that will cause it to change its state of matter.
This room is called the hallway. In it, you see:
a picture
a substance called air
the agent
You also see:
A door to the green house (that is open)
A door to the living room (that is open)
A door to the art studio (that is open)
A door to the kitchen (that is open)
A door to the bedroom (that is open)
A door to the workshop (that is open)

## D.4 HYPERPARAMETERS

Below we list the key hyperparameters required for both OPC-SFT and reinforcement learning.

Table 10: OPC-SFT hyperparameters for **ALFWorld** and **ScienceWorld**.

| Hyperparameter | ALFWorld | ScienceWorld |
|---|---|---|
| Clipping ratio $\epsilon$ | 0.5 | 0.5 |
| Learning rate | 1e-5 | 2e-6 |
| Rollout batch size | 256 | 256 |
| Train batch size | 32 | 32 |
| Maximum epochs | 3 | 3 |
| Number of episodes | 3 | 3 |
| Prompt maximum length | 4000 | 4000 |

Table 11: Reinforcement learning hyperparameters for **ALFWorld** and **ScienceWorld**.

| Hyperparameter | ALFWorld | ScienceWorld |
|---|---|---|
| Learning rate | 1e-6 | 1e-6 |
| KL loss coefficient | 0.01 | 0.01 |
| KL coefficient | 0.001 | 0.01 |
| KL loss type | Low Var KL | Low Var KL |
| Rollout temperature | 0.7 | 0.7 |
| Validation temperature | 0.7 | 0.7 |
| Maximum prompt length | 8192 | 8192 |
| Maximum response length | 256 | 128 |
| Clipping ratio low | 0.2 | 0.2 |
| Clipping ratio high | 0.2 | 0.2 |
| Rollout N | 8 | 8 |
| Max environment steps | 40 | 30 |
| PPO mini batch size | 16 | 32 |
| Max number of sequences | 512 | 1024 |
| Critic warm-up | 0 | 0 |

## D.5 OUT-OF-DISTRIBUTION DATASETS

To provide a comprehensive evaluation of knowledge retention and generalization, we include several widely used benchmarks from different domains. Below we briefly describe each dataset:

**MBPP (Austin et al., 2021).** The Mostly Basic Python Problems (MBPP) dataset consists of 378 hand-written Python programming problems designed to evaluate models' ability to generate correct and efficient code. Each problem includes a description and a reference implementation, and performance is measured using functional correctness tests.

**HumanEval (Chen et al., 2021).** The HumanEval dataset provides 164 Python programming tasks accompanied by unit tests. It is commonly used to assess the code generation ability of large language models.

**MMLU (Hendrycks et al., 2021a).** The Massive Multitask Language Understanding (MMLU) benchmark evaluates broad general knowledge across 57 tasks covering mathematics, history, law, medicine, and other domains. It is designed to test both world knowledge and problem-solving ability.

**GPQA (Rein et al., 2023).** The Graduate-Level Google-Proof Q&A benchmark contains 198 challenging questions curated by subject matter experts, with a focus on requiring reasoning beyond simple retrieval.

**LiveCodeBench (Jain et al., 2025)** The LiveCodeBench benchmark evaluates live code generation and execution ability under dynamic environments. It provides 442 diverse programming challenges with runtime validation, extending beyond static unit-test benchmarks.

