# OpenReview forum: "Off-Policy Token Clipped Supervised Fine-Tuning Yields a Robust Cold-Start"
_ICLR.cc/2026/Conference — Submitted to ICLR 2026_

### Official Review · Reviewer_N85X · 2025-10-27

**Soundness:** 3
**Presentation:** 2
**Contribution:** 2
**Rating:** 6
**Confidence:** 3

**Summary:**

This paper identifies a primary cause of **catastrophic forgetting** during the Supervised Fine-Tuning (SFT) of Large Language Models (LLMs). The authors argue that SFT, often used as a "cold-start" for Reinforcement Learning (RL), is destabilized by what they term **"off-policy" tokens**—tokens within the expert data to which the base model assigns a very low probability. They demonstrate that the standard log-likelihood objective assigns disproportionately **large gradient magnitudes** to these tokens, causing significant representational drift and the erosion of pre-trained knowledge, particularly in the initial stages of training.

To address this instability, the paper proposes **Off-Policy Token-Clipped SFT (OPC-SFT)**. This method adapts the clipping mechanism from the Proximal Policy Optimization (PPO) algorithm, a trust region method used in RL. OPC-SFT moderates the learning process by constraining gradient updates from off-policy tokens. It achieves this by clipping the token-level probability ratio to a bounded interval. The primary contributions include identifying this "off-policy" token problem and proposing the OPC-SFT solution. Experiments on the ALFWorld and Science World agentic benchmarks show that this approach **reduces OOD forgetting by 11.54%** and **improves final RL performance by 6.70%**. This is supported by latent-space analysis, which shows that OPC-SFT results in less representational drift.

**Strengths:**

The main strength of this paper is its original, theoretically grounded approach to addressing catastrophic forgetting during Supervised Fine-Tuning (SFT). The authors offer a well-supported diagnosis: low-probability “off-policy tokens” in expert data produce disproportionately large gradients, which destabilize the model’s internal representations. The evidence presented for this claim is convincing.

The solution, Off-Policy Token-Clipped SFT (**OPC-SFT**), is an adaptation of the PPO clipping mechanism from Reinforcement Learning (RL) to the supervised setting. While not particularly novel, the clipping immediately has practical significance, demonstrating a substantial average 11.54% reduction in out-of-distribution forgetting and a significant 6.70% increase in final RL agent performance across challenging agentic benchmarks, such as ALFWorld and Science World (for Qwen2.5 and Llama 3).

Furthermore, the paper is commended for its high quality and clarity in technical validation. The methodology is rigorously tested against strong baselines (including DFT and NEFTune) across three different model backbones, ensuring the generality of the results. The inclusion of comprehensive diagnostic analyses—specifically the latent-space shift visualization via PCA—provides clear, mechanistic evidence that OPC-SFT successfully preserves the model's prior knowledge by limiting representational drift. The presentation of the research is exceptionally clear, logically progressing from problem identification (large gradient norms in early SFT) to the derived solution ($\mathcal{L}_{OPC-SFT}$), making the core concepts and the derivation of the novel loss function highly accessible to readers.

**Weaknesses:**

The paper's primary weakness lies in the justification and novelty of its core mechanism. The authors are transparent in citing concurrent work (Zhu et al., 2025) that also explores applying a PPO-style clipping mechanism to SFT. This significantly narrows the paper's novel contribution to its analysis—framing the problem as one of "off-policy tokens" causing catastrophic forgetting—rather than a novel algorithmic solution. Furthermore, the adaptation from the PPO objective to SFT involves a critical simplification that is not adequately justified: setting the advantage function $\hat{A}=1$ (Eq. 8-9). This assumes every token in the expert data is equally and maximally "good." This is a strong and likely incorrect assumption; it conflates mundane tokens (e.g., "the," "is") with critical reasoning tokens and ignores any potential sub-optimality in the expert data. The paper would be much stronger if it explored or justified this choice, for example by ablating against alternative, non-uniform advantage weightings (e.g., weighting by token surprisal, or weighting down tokens the base model already knows well).

**Questions:**

The adaptation from the PPO objective to the proposed $\mathcal{L}_{OPC-SFT}$ (Eq. 9) hinges on the critical simplification of setting the advantage $\hat{A}=1$. This assumes all tokens in the expert data are equally and maximally optimal, which seems unlikely (e.g., it equates mundane tokens like "the" with critical, task-specific tokens). Could the authors please provide a more detailed justification for this choice? Furthermore, have you experimented with alternative, non-uniform advantage weightings? For example, weighting tokens by their surprisal (negative log-probability) under the base model, or down-weighting tokens the model already assigns high probability to. This would help clarify if this simple, uniform weighting is a robust choice or a missed opportunity for a more nuanced credit assignment.

The empirical evidence shows clear success on agentic tasks (ALFWorld, ScienceWorld), which the paper effectively argues are highly "off-policy." However, the gains are admittedly "modest" on math tasks, which are shown to be more "on-policy" (Fig. 4). This suggests the method's utility might be a niche solution for specific domain adaptation rather than a general-purpose SFT improvement. To substantiate the paper's broader claims, could the authors provide any results or analysis on how OPC-SFT performs on standard, large-scale instruction-tuning benchmarks (e.g., a subset of OpenOrca or Alpaca)? This would help clarify if the method provides benefits when the fine-tuning data is more diverse and "closer-to-policy."

The stability and performance of trust-region methods heavily depend on how the reference policy ($\pi_{\theta_{old}}$) is managed. The paper states this is "periodically refreshed" (Sec 3.3) but provides no details on this crucial hyperparameter. Could the authors please specify what update frequency was used for $\pi_{\theta_{old}}$ in the experiments (e.g., updated every N steps, every epoch)? More importantly, could you provide an ablation study or analysis showing how sensitive the model's performance is to this update frequency?

---

> ### Author Response · Authors · 2025-11-25
>
> Dear Reviewer N85X,
>
> Thank you for your insightful comments. We sincerely hope that our rebuttal could properly address your concerns.
>
> **Weakness 1. Novelty of our core mechanism.**
>
> We fully understand the concern. We would like to clarify that our primary contribution lies in identifying the underlying cause of catastrophic forgetting during SFT: low-probability tokens that induce disproportionately large gradient norms. This finding is non-trivial, as it provides the key insight that motivates our method and explains why certain updates are harmful in SFT. This also implies that our motivation differs from concurrent work, and so does our empirical focus. Beyond mathematical reasoning, we show significant improvements on agentic tasks. This broader evaluation highlights the practical significance of our analysis and method.
>
> Regarding the clipping mechanism, we agree that it is not a novel algorithmic invention. As noted in the paper, it is a commonly adopted trick and may indeed be considered concurrently by other researchers. However, we hold that this does not diminish our contribution, rather, it highlights that clipping is a broadly useful technique whose effectiveness becomes clearer when grounded in our analysis of forgetting dynamics.
>
> In addition, we compare our method with other trust-region approaches like KL penalty and find that the critical factor is the selective, token-level clipping, rather than merely constraining the global divergence between consecutive policies. This distinction is essential to understanding our motivation that off-policy tokens lead to the forgetting of LLM.
>
> **Weakness 2 & Question 1. How to set the advantage when adapted from PPO?**
>
> Thanks for your question. Our motivation is that low-probability tokens during SFT cause forgetting, which is why we clip their updates. Thus, our reference to PPO is solely to introduce the clipping mechanism, not to adapt PPO to the SFT setting. Our framework remains grounded in the Maximum Likelihood Estimation (MLE) setting, which is effectively supervised learning. Equation 9 can be viewed as a generalized supervised learning objective. Consequently, we can directly follow the rule in vanilla SFT, where all weights are uniform, as standard supervised learning does not prioritize specific tokens. While our method introduces implicit weighting via the clipping ratio, we compared it against the vanilla baseline. We found that our approach yields greater in-distribution ability while significantly reducing the impairment of general abilities, which means less catastrophic forgetting compared to the vanilla method.
>
> **Question 2. Add the instruction-tuning benchmark.**
>
> We thank the reviewer for this constructive suggestion. We fully understand the concern that OPC-SFT might appear tailored for 'off-policy' agentic tasks, and we agree that demonstrating its utility on general instruction tuning is crucial.
> To address this, we extended our evaluation to standard, large-scale instruction-tuning benchmarks. Due to time constraints during the rebuttal period, we sampled a 60k subset from the OpenOrca dataset to fine-tune Qwen2.5-7B and Llama3.1-8B. After supervised learning, we evaluated 'in-distribution' generalization using AlpacaEval 2.0 and 'out-of-distribution' robustness to measure catastrophic forgetting using code benchmarks (HumanEval and MBPP).
> As shown in the table above, OPC-SFT consistently outperforms standard SFT on out-of-distribution code tasks across both models, demonstrating that our method significantly mitigates catastrophic forgetting of general abilities while learning new instructions. Furthermore, on AlpacaEval 2.0, OPC-SFT improves the Length-Controlled Win Rate for both models. These results confirm that OPC-SFT is not a specialized solution but rather a widely applicable method that enhances general instruction-following capabilities while preserving the model's prior knowledge better than standard SFT.
>
> | | **In-distribution** | | | **Out-of-distribution** | |
> | :--- | :---: | :---: | :---: | :---: | :---: |
> | **Method** | **LC win rate** | **Win rate** | | **HumanEval** | **MBPP** |
> | **Qwen2.5-7B SFT** | 11.9 | 8.1 | | 58.0 | 31.7 |
> | **Qwen2.5-7B OPC-SFT** | **12.9** | **11.4** | | **63.5** | **34.1** |
> | **Llama3.1-8B SFT** | 14.3 | **13.5** | | 48.0 | 35.7 |
> | **Llama3.1-8B OPC-SFT**| **14.5** | 12.1 | | **49.1** | **37.3** |

---

> ### Author Response · Authors · 2025-11-25
>
> **Question 3. How the reference policy is managed.**
>
> We sincerely apologize for not clearly explaining how the policy $\pi_{\theta_{old}}$ is maintained in our method. To avoid misunderstanding, we have revised the term 'reference policy' to 'old policy' that updates, which is updated after each training step of the policy model on 256 samples. Specifically, we reset $\pi_{\theta_{old}}$ after every update batch of 256 samples, which corresponds to one actor training step on 256 trajectories. We refer to this as the old policy update frequency. We completely agree that this frequency cannot be too large or too small, and that an ablation study on how the old policy is managed is necessary. Hence, we conducted an ablation study by varying the old policy update frequency, including updating every 128, 512, and 1024 samples. The results show that overly infrequent updates lead to excessive clipping, whereas moderate update frequencies, like 256 samples, achieve stable training. We have also added Sec.4.5 for this ablation study in the updated manuscript.
>
>
> | Old Policy Update Freq | Before RL ALFWorld-seen | Before RL ALFWorld-unseen |
> |------------------------|-------------------------|----------------------------|
> | 128                    | 77.14                   | 76.87                      |
> | 256                    | 78.57                   | 79.10                      |
> | 512                    | 75.00                   | 75.37                      |
> | 1024                   | 73.57                   | 73.88                      |
>
> | Old Policy Update Freq | After RL ALFWorld-seen | After RL ALFWorld-unseen |
> |------------------------|------------------------|---------------------------|
> | 128                    | 93.27                  | 91.79                     |
> | 256                    | 94.29                  | 92.54                     |
> | 512                    | 93.57                  | 90.30                     |
> | 1024                   | 89.29                  | 88.81                     |
>
> | Old Policy Update Freq | MBPP  | MMLU  | HumanEval | GPQA  | LiveCodeBench | MATH500 |
> |------------------------|-------|-------|-----------|-------|---------------|---------|
> | 128                    | 58.99 | 57.84 | 47.98     | 17.17 | 36.65         | 35.60   |
> | 256                    | 58.71 | 59.97 | 48.13     | 18.18 | 42.60         | 37.60   |
> | 512                    | 61.11 | 57.03 | 48.89     | 19.70 | 36.79         | 36.20   |
> | 1024                   | 60.05 | 59.26 | 46.53     | 21.21 | 35.68         | 36.40   |

---

### Official Review · Reviewer_VFg9 · 2025-11-01

**Soundness:** 3
**Presentation:** 2
**Contribution:** 2
**Rating:** 4
**Confidence:** 4

**Summary:**

This work identifies a limitation in the standard SFT-before-RL training paradigm:  overfitting and catastrophic forgetting caused by large gradients from "off-policy" (low-probability) tokens in SFT data.
To mitigate this, the authors propose OPC-SFT, a token-level clipping mechanism inspired by the clipping operation in PPO-style RL objectives.
Experiments on the ALFWorld and ScienceWorld benchmarks suggest that OPC-SFT reduces forgetting on out-of-distribution tasks and improves subsequent RL performance, compared to standard SFT and other baselines.

**Strengths:**

* Provides a clear empirical analysis of the identified problem, supported by relevant evidence.
* Proposes a simple, practical method that is easy to implement.
* Experimental evaluation spans diverse tasks and models, showing consistent improvements.
* Intermediate diagnostic metrics (e.g., principal components, gradient norm distributions) yield additional insights, e.g., explain why agentic tasks are more affected than math/reasoning tasks.

**Weaknesses:**

**Issue 1: Limited clarity on the rationale behind the proposed method.**
- There is a discrepency between the analysis and the proposed clipping mechanism.
  - The analysis defines *off-policy tokens* as those with small model probability $\pi_{\theta}$, which seems to align more with the DFT method that weights token loss by token probability.
  - The proposed OPC-SFT method instead clips tokens with a large probability ratio $\pi_{\theta} / \pi_{\text{old}} > 1 + \epsilon$, i.e., tokens whose probability increased substantially after a few policy update steps.
- In Section 3.3, the explanation connecting OPC-SFT to PPO’s clipping mechanism is tenuous and potentially misleading. One major discrepency is that PPO’s rationale fundamentally relies on an on-policy sampling assumption and the policy gradient theorem, which are totally irrelevant for SFT.
- The final loss in Eq. (9) for OPC-SFT essentially applies $\min\\{r(\theta), 1 + \epsilon\\}$, meaning *a token's gradient is clipped if and only if its probability ratio exceeds $1 + \epsilon$*. That's all. An intuitive, accurate and straightforward explanation that could save pages of analysis in Section 3.2, 3.3 and Appendix A. I would suggest that the authors consider reframing and clearly articulating the actual rationale behind this choice.
- The argument in Lines 205–209 is unconvincing: a small denominator in Eq. (5) (which can be rewritten as $\nabla_{\theta} \log \pi_{\theta}$) does not inherently imply a large gradient norm.



**Issue 2: Insufficient analysis and discussion of critical hyperparameters.**
- The refresh period for $\pi_{\text{old}}$ is an important hyperparameter, yet it is barely discussed (only briefly mentioned in Line 245).
  - If the period is 1, clipping is never triggered, and the proposed OPC-SFT method reduces to standard SFT.
  - If the period is too large, clipping can occur for nearly all tokens, potentially halting learning.
- A detailed analysis or ablation study on the effect of this parameter, alongside $\epsilon$, would significantly strengthen the paper.

**Questions:**

- Line 74: How exactly are the 11.54% and 6.70% numbers calculated?
- Figure 1 (c): What does “episode” refer to here? Should it be “epoch”?
- Table 1: Why does the Qwen 1.5B model outperform the 7B model significantly? Similar question about Table 3.
- References to Tab. B.1 (Line 761) and Tab. B.2 (Line 793) are unclear, maybe typos?
- Line 1204: The rollout temperature for RL is set to 0.7, smaller than the commonly used value of 1. Is there a specific motivation for this choice?

---

> ### Author Response · Authors · 2025-11-25
>
> Dear Reviewer VFg9,
>
> Thank you for your insightful comments. We sincerely hope that our rebuttal could properly address your concerns.
>
> **Weakness 1. Discrepancy between the analysis and proposed method.**
>
> Thank you for your time. We would like to clarify the relationship between our low-probability token analysis and the proposed method. Firstly, let us explain our motivation. As shown in our added analysis https://i.postimg.cc/VkJZx1Vz/gradient-bins-two-axes2-01.png, we find that low-probability tokens indeed induce large gradient norms, which in turn cause large probability shifts and lead to forgetting of pre-trained knowledge. Hence, we aim to constrain updates on these low-probability tokens to preserve prior knowledge and prevent forgetting. Trust region methods naturally provide a proximal optimization mechanism for limiting the magnitude of policy updates using an old policy $\pi_{\theta_{\text{old}}}$. Regarding token-level clipping, we introduce the ratio between the current policy and the old policy, and clip the update if the ratio exceeds the threshold. In this way, low-probability tokens are given less update space. This is why we introduce the policy ratio based on our findings. Specifically, we apologize for not clearly explaining the role of $\pi_{\theta_{\text{old}}}$ in the earlier version. In our method, $\pi_{\theta_{\text{old}}}$ is updated at every rollout batch and is introduced to enforce proximal updates.
>
> Besides clipping, we also notice another line of approaches which applies a KL-Divergence penalty. It measures the discrepancy between the two policies at the batch-level, and do not show the direct connection to token-level update constraint for low-probability tokens. Also, it requires tuning a difficult hyperparameter. The clipping strategy, however, operates directly at the token level and explicitly provides less space for optimizing off-policy tokens, making it particularly simple yet effective at suppressing excessive updates to these tokens. Therefore, we adopt the clipping mechanism for its stability and simplicity. Thus, our adopted clipping strategy using the probability ratio is fully aligned with our motivation to mitigate forgetting induced by low-probability tokens.
>
> **Weakness 2. Redundant content and misleading relationship between OPC-SFT and PPO.**
>
> Thanks for your suggestion, and we have revised the manuscript accordingly. Our motivation is that low-probability tokens during SFT cause forgetting, which is why we clip their updates. Our initial intention to connect with PPO was only to highlight an operational similarity, specifically, the use of a clipping rule, not to rely on PPO’s on-policy assumptions or policy-gradient-theorem rationale, which are indeed irrelevant for SFT.
> In our setting, the rationale is much simpler: to prevent low-probability tokens that induce large gradient norms from reshaping the model parameters and causing forgetting. Clipping based on the probability ratio provides a direct and effective way to stop those harmful updates, and this is the sole reason we adopt the ratio form. We have reorganized Sections 3.2 and 3.3 to more clearly articulate this intuition.
> The revised version now focuses on two key points: (1) our empirical finding that low-probability tokens produce disproportionately large gradients that lead to forgetting of general abilities, and (2) how a commonly used clipping trick can be applied in a straightforward manner to mitigate this issue. We have also significantly shortened the discussion to avoid earlier conceptual tangling.
>
> **Weakness 3. Probability and gradient of tokens.**
>
> We fully understand your concern. Our earlier statement was not intended as a rigorous mathematical proof, and we agree that a small denominator in Eq. (5) does not inherently imply a large gradient norm. To address this, we have added additional empirical evidence. Specifically, we include an empirical analysis showing the distribution of gradient norms across different token-probability bins. It is easy to observe that the mean gradient norm decreases as the token probability increases, as illustrated in https://i.postimg.cc/VkJZx1Vz/gradient-bins-two-axes2-01.png, which is now included in Sec.3.2 of the revised manuscript.

---

> ### Author Response · Authors · 2025-11-25
>
> **Weakness 4. Insufficient analysis and discussion of critical hyperparameters.**
>
> We sincerely apologize for not clearly explaining how the policy $\pi_{\theta_{old}}$ is maintained in our method. To avoid misunderstanding, we have revised the term 'reference policy' to 'old policy' that updates, which is updated after each training step of the policy model on 256 samples. Specifically, we reset $\pi_{\theta_{old}}$ after every update batch of 256 samples, which corresponds to one actor training step on 256 trajectories. We refer to this as the old policy update frequency. We completely agree that this frequency cannot be too large or too small. To further evaluate this design choice, we conducted an ablation study by varying the old policy update frequency, including updating every 128, 512, and 1024 samples. The results show that overly infrequent updates lead to excessive clipping, whereas moderate update frequencies, like 256 samples, achieve stable training. We have also added Section 4.5 for this ablation study in the updated manuscript.
>
> | Old Policy Update Freq | Before RL ALFWorld-seen | Before RL ALFWorld-unseen |
> |------------------------|-------------------------|----------------------------|
> | 128                    | 77.14                   | 76.87                      |
> | 256                    | 78.57                   | 79.10                      |
> | 512                    | 75.00                   | 75.37                      |
> | 1024                   | 73.57                   | 73.88                      |
>
>
> | Old Policy Update Freq | After RL ALFWorld-seen | After RL ALFWorld-unseen |
> |------------------------|------------------------|---------------------------|
> | 128                    | 93.27                  | 91.79                     |
> | 256                    | 94.29                  | 92.54                     |
> | 512                    | 93.57                  | 90.30                     |
> | 1024                   | 89.29                  | 88.81                     |
>
>
> | Old Policy Update Freq | MBPP  | MMLU  | HumanEval | GPQA  | LiveCodeBench | MATH500 |
> |------------------------|-------|-------|-----------|-------|---------------|---------|
> | 128                    | 58.99 | 57.84 | 47.98     | 17.17 | 36.65         | 35.60   |
> | 256                    | 58.71 | 59.97 | 48.13     | 18.18 | 42.60         | 37.60   |
> | 512                    | 61.11 | 57.03 | 48.89     | 19.70 | 36.79         | 36.20   |
> | 1024                   | 60.05 | 59.26 | 46.53     | 21.21 | 35.68         | 36.40   |
>
>
> **Question 1. The performance advantage calculation.**
>
> Thank you for this question. We apologize that the calculation method was not explicitly detailed in the initial text. We have clarified these statistics in the revised manuscript as follows:
> The Advantage of OOD Performance
> The 11.54% advantage represents the mean OOD performance gain of OPC-SFT over SFT. It is calculated by averaging the six individual percentage gains across the three models and two OOD settings (Tables 2 & 7).
>
> Individual Gains (ALFWorld & ScienceWorld): +8.71%, +9.96%, +17.48%, and +10.24%, +12.85%, +9.99%
>
> Final Calculation: $(8.71 + 9.96 + 17.48 + 10.24 + 12.85 + 9.99)/6 = 11.54.$To improve visibility, we have moved Table 7 from the appendix of the original submission to the main text, where it is now labeled as Table 3.
>
> The RL Performance Advantage
>
> We apologize for the earlier confusion. Our original number was computed in a tangled, task-level manner across different models to compare OPC-SFT with SFT. We have now revised it to 7.09%, which is straightforward to verify. We realized that calculating the improvement directly from the `Average' columns in Table 3 is more transparent and allows readers to easily check the result, representing the average relative improvement across the three backbones. The calculation is shown as follows:
>
> Qwen2.5-7B: 77.70% vs. 69.97% $\rightarrow$ +11.05% advantage
>
> Qwen2.5-1.5B: 79.55% vs. 74.27% $\rightarrow$ +7.10% advantage
>
> Llama3.2-3B: 81.49% vs. 79.02% $\rightarrow$ +3.12% advantage
>
> Final Average Advantage: $(11.05 + 7.10 + 3.12) / 3 = 7.09%$
> We believe this calculation provides a clear and robust measure of our method's superiority.  We hope this calculation detail will ensure clarity.

---

> ### Author Response · Authors · 2025-11-25
>
> **Question 2. Figure 1 (c): What does 'episode' refer to here? Should it be 'epoch'?**
>
> We apologize for not clearly explaining the axis in Figure 1(c). The label 'episode' indeed means 'epoch'. In our setting, one episode corresponds to training on the entire dataset once, which is equivalent to one epoch. And one epoch consists of 15 gradient steps. In the revised manuscript, we have updated the axis labeling in the figure to avoid confusion.
>
> **Question 3. Why 1.5B outperforms 7B?**
>
> Thanks for the observation. We reported the actual experimental findings. We hypothesize that the unified hyperparameters we use across both models may have caused the 7B model to converge to an unfavorable internal state during training, but are beneficial for the training of the 1.5B model. Such counter-intuitive results have also been observed in prior related work [1], [2], both in math-reasoning tasks and agentic settings.
>
> [1] Divide and Conquer: Grounding LLMs as Efficient Decision-Making Agents via Offline Hierarchical Reinforcement Learning. ICML 2025.
>
> [2] Group-in-Group Policy Optimization for LLM Agent Training. NeurIPS 2025.
>
> **Question 4. References typos.**
>
> Thanks for pointing out the problem. We have changed it to the updated version.
>
> **Question 5. The rollout temperature.**
>
> Thanks for your suggestion. We have noticed that recent works [1], [2], [3], in the reinforcement learning phase during post-training, also adopt a temperature of 0.7. Moreover, we have also tested rollout with temperature = 1.0, and the results show no significant difference.
>
>
> | Method                     | ALFWorld-seen | ALFWorld-unseen | ScienceWorld-seen | ScienceWorld-unseen |
> |----------------------------|---------------|-----------------|-------------------|---------------------|
> | Rollout Temperature=0.5    | **92.14**     | **91.04**       | 66.49             | 61.14               |
> | Rollout Temperature=1      | 89.29         | 87.31           | **70.10**         | **66.35**           |
>
> [1] Language Models that Think, Chat Better. CoRR, abs/2509.20357, 2025.
>
> [2] DeepSeek-R1 incentivizes reasoning in LLMs through reinforcement learning. Nature 2025.
>
> [3] Unpacking DPO and PPO: Disentangling Best Practices for Learning from Preference Feedback. NeurIPS 2024

---

### Official Review · Reviewer_tn6h · 2025-11-02

**Soundness:** 2
**Presentation:** 2
**Contribution:** 2
**Rating:** 2
**Confidence:** 3

**Summary:**

The paper investigates catastrophic forgetting in LLM models, and particularly why SFT can lead to a higher degree of forgetting compared to RLFT. The work argues that this is mostly caused by "off-policy" token in SFT which leads to large magnitude gradients and therefore forgetting. To resolve this problem the authors propose to borrow the clipping machinery used by the PPO algorithm, and show improved memory retention and act as a better cold-start for the RL finetuning stage.

**Strengths:**

The paper identifies a meaningful problem in the finetuning pipeline of LLMs, and provides a simple yet effective solution of importing clipping gradient strategy from PPO. The authors show empirically the efficacy of their approach. As far as I can tell, the proposed solution, relying on clipping is novel and interesting.

**Weaknesses:**

While the paper is generally clear and easy to read there are a few issues around contextualizing the work that I would like to point out.
First I'm not convinced by the terminology used, namely that of "off-policy tokens" jointly with supervised learning. I understand that you can view SFT as imitation learning, I just find the abuse of language excessive, and potentially leading to confusions later. In my opinion I would call either both stages RL finetuning stages, just that the first one is off-policy imitation learning one. Or call the first stage supervised finetuning, but then use the language of probabilistic modelling to describe what you are doing, not that of RL, which is sufficiently expressive. This is also potentially the paper to be more explicit or formal about the proposed method.

The other weakness that I find in the text is that it does not connect at all with continual learning, a subfield that has studied catastrophic forgetting extensively, bot the causes of its emergence in neural networks as well as providing several potential solutions. This disconnect to a large body of work seems concerning. I would suggest at least citing some survey (they are several) in this space and acknowledging the existence of the field.

Going into the specifics. The paper claims that large magnitude gradients resulting from low probability tokens during the SFT finetuning are the cause of forgetting. While I do not completely disagree with the hypothesis, I feel like further evidence could be useful. Is the claim that from an optimization point of view this "erratic" gradients (line 208) are making optimization unstable (e.g. we are in a high curvature region). And that is why they are disruptive?
To reframe my question maybe. I do not think the norm of the gradient is the problem (or at least the cause). The actual problem is that when learning something new, not having access to the previous task that we are attempting not to forget, the best course of action is to be conservative and ask to stay as close as possible to the previous policy (under some metric of choice). Which is exactly what PPO does as well (stay close to pi_old). So is not the dynamics of the learning process (i.e. the magnitude of the gradient) that is the problem, is that we are balancing how much we want to learn the new task vs attempt to preserve knowledge by being very conservative and saying do not change anything.

I'm bringing this up because the dynamics and justifications in PPO are a bit different, and the clipping (or more formally the KL term that the clipping is meant to approximate) is a term to induce stability in training and make sure PPO does not diverge. However SFT does not exhibit any kind of divergence behavior (even if gradients are large), therefore I believe the motivation has to be different. I find the current framing to rely on this idea of unstable learning in SFT which I believe to be misleading.

Following of this, I feel like equation (6) is not sufficiently justified. I know the authors point to concurrent work, but by naming the work concurrent it means that the formulation is novel as with respect to this paper, and therefore it should be justified (rather than rely on the other paper as justification). I feel this is where the SFT nomenclature becomes fuzzy. How does equation 6 relate to equation 1. How can we understand this from the MLE point of view? Should we still call this SFT then? Or maybe if not equation 6 then equation 9. Would it had been better to derive this from a constraint optimization perspective, similar to PPO via a KL term on the difference between the old and new distribution?  Also the constraint (trust region view) would then directly connect with many CL algorithms (at least regularization based ones), which take the form of a constraint optimization that minimizes the loss on the current task under a KL constraint with respect to the old task, with the difference from many of these works that here the KL constraint is being approximate via a clipping strategy akin to PPO.

Of course the authors do not need to fully agree with my perspective, but I find the derivation in section 3.3 very hand-wavy, relying on similarities rather than a more formal derivation. Which can lead to question of what is the objective in (9) actually minimizes ? What will it converge to? etc.

Empirical results look good, but I was wondering if it would make sense to have a baseline that involves replay during the SFT phase? Or argument against it in the text.

**Questions:**

1. Can you justify the semantics of objective in equation 9 as a supervised objective ? What would be the correct way of interpreting it?
2. Can you provide some more insights of whether is the gradient magnitude that is problematic or something different? Why would then approaches like imposing gradient clipping (used for e.g. often when you dealing with exploding gradients), or decreasing learning rate be a solution?
3. Can you provide any vanilla standard continual learning baseline for the experiments (e.g. replay as the most simple one to implement).

**Details Of Ethics Concerns:**

I do not see reasons for an ethics review.

---

> ### Author Response · Authors · 2025-11-25
>
> Dear Reviewer tn6h,
>
> Thank you for your insightful comments. We sincerely hope that our rebuttal could properly address your concerns.
>
> **Weakness 1. The abuse of terminology.**
>
> Thank you for your review. Our motivation is that low-probability tokens during SFT cause forgetting, which is why we clip their updates. However, the terminology used has caused some misunderstanding, so we have reconsidered our use of the term 'off-policy tokens'. What we intend to convey by 'off-policy' is the divergence or distance of the token in the data from the policy's behavior. This terminology originates from reinforcement learning, and we use it to draw an analogy to off-policy learning: the data being learned from is not aligned with the model’s current behavior, and the corresponding target tokens receive a low probability under the model. The term is used purely for explanatory purposes.
>
> We appreciate your suggestion regarding renaming. We have explored the option of framing the first stage as imitation learning, but we found that this terminology is less common and less precise in the LLM post-training literature. Alternatively, if we refer to the first stage strictly as supervised fine-tuning, then we would need to fully adopt the probabilistic-modelling perspective and avoid RL terminology. In that case, we would like to confirm whether expressions such as `low-probability' tokens, as used in [1], remain appropriate for describing our observations. If you believe a different phrasing would better avoid confusion, we would be very happy to adopt your recommended terminology.
>
> As a clarification, we use 'off-policy' only to express a similar intuition as in RL, and we explicitly explain the borrowed meaning in the revised manuscript introduction. Additionally, recent work [2] increasingly views SFT and RL as part of a unified post-training pipeline, so we would like to ask whether this terminology is acceptable in that broader context. We sincerely appreciate your guidance on this point.
>
> [1] Do Not Let Low-Probability Tokens Over-Dominate in RL for LLMs. CoRR, abs/2505.12929, 2025.
>
> [2] Towards a Unified View of Large Language Model Post-Training. CoRR, abs/2509.04419, 2025.
>
> **Weakness 2 & 3. Motivation is lacking evidence and in the text this paper does not connect at all with continual learning.**
>
> We fully understand your concern. Firstly, we would like to clarify our motivation and provide additional experimental analysis. Then, we will discuss the connection and differences with continual learning.
>
> Our motivation stems from the observation that low-probability tokens indeed induce large gradient norms, which in turn cause significant probability shifts and lead to the forgetting of pre-trained knowledge. This is why we clip their updates. We agree with the reviewer’s perspective that when the model encounters unfamiliar tokens, the optimization process naturally requires substantial parameter updates. However, our empirical findings reveal a critical trade-off. These aggressive updates often create irreversible shifts in the model’s internal representations, which degrade its performance on pre-trained tasks. Since the model’s general abilities arise from an extremely diverse and abundant pre-training dataset, our goal is not to replay this data, but rather to ensure the model learns new tasks progressively, minimizing the cost to previously acquired knowledge.
>
> Additionally, we have added experiments that provide further evidence showing that large probability shifts strongly correlate with high gradient norms, as demonstrated in  https://i.postimg.cc/VkJZx1Vz/gradient-bins-two-axes2-01.png.
>
> Regarding the terms 'disruptive' and 'erratic', we apologize for the confusion and have removed them. Our initial intention was to describe the negative impact these updates can have on the model's general capabilities.
>
> Consequently, our approach focuses on selectively clipping updates associated with high-magnitude gradients to preserve prior knowledge learned during the pre-training stage on abundant data, which is crucial for maintaining the model's general abilities in subsequent RL optimization.

---

> ### Author Response · Authors · 2025-11-25
>
> Regarding continual learning (CL), we apologize for not addressing this important area of research earlier, and we have added a discussion of CL in the revised manuscript, specifically in the Appx.C.3 "Related Work" section, along with representative citations.
>
> However, it is important to clarify the conceptual difference between CL and our setting. CL typically considers a multi-task or sequential-task scenario, where the model must retain performance on earlier tasks while learning new ones. In contrast, our work focuses on preserving general pre-trained knowledge when post-training on a comparatively small dataset. Given that the pre-training corpus contains a vast and diverse set of data compared to the much smaller SFT dataset, designing an effective and representative replay buffer for preserving general abilities is non-trivial. For example, although replaying DeepScaleR helps the model retain its math-reasoning ability, it still shows limited performance on coding tasks. This illustrates why standard CL formulations may not directly apply to our setting.
>
> We agree that CL has developed a rich body of work on mitigating forgetting, including replay-based approaches, which are particularly effective when a model learns a sequence of tasks. Since these methods also aim at preventing forgetting, even if it's on previous tasks, we have included CL baselines. First, we have incorporated a replay-based method as a baseline when tuning on agentic tasks, where we replay the math dataset DeepScaleR. Second, we note that KL penalty trust-region constraints can be viewed as a form of regularization, analogous to CL techniques. For the KL penalty baseline, since computing the exact KL divergence is not efficient, we use the empirical expectation over the offline dataset. The results are presented below.
>
>
> | Method      | Before RL ALFWorld-seen | Before RL ALFWorld-unseen | After RL ALFWorld-seen | After RL ALFWorld-unseen |
> |-------------|-------------------------|---------------------------|------------------------|--------------------------|
> | KL penalty  | 72.14                   | 73.13                     | 87.86                  | 82.84                    |
> | Replay      | 71.43                   | 77.61                     | 83.57                  | 76.12                    |
> | OPC-SFT     | 76.43                   | 77.61                     | 94.29                  | 92.54                    |
>
>
> | Method      | MBPP  | MMLU  | HumanEval | GPQA  | LiveCodeBench | MATH500 |
> |-------------|-------|-------|-----------|-------|---------------|---------|
> | KL penalty  | 56.61 | 58.12 | 48.70     | 16.16 | 36.83         | 36.20   |
> | Replay      | 55.82 | 61.92 | 46.57     | 16.67 | 36.00         | 46.80   |
> | OPC-SFT     | 58.71 | 59.97 | 48.13     | 18.18 | 42.60         | 37.60   |
>
> Due to time and resource limitations during the rebuttal stage, we currently report results only on Llama3.2-3B-Instruct, and we will include results for all models in the final version.
>
>
> **Weakness 4. The connection between PPO is misleading.**
>
> We sincerely apologize for the confusion. Our motivation is that low-probability tokens during SFT cause forgetting, which is why we clip their updates. Thus, our reference to PPO is solely to introduce the clipping mechanism, which can be used for proximal optimization. The main problem we aim to address is forgetting during SFT when it is performed on a small dataset for cold-start tasks. Therefore, our motivation differs from the stability issues that PPO is designed to resolve. While our method of adopting clipping is inspired by PPO, the inspiration lies in its similarity to proximal policy optimization: we clip tokens that would otherwise induce large gradient norms, which significantly reshape the model parameters and ultimately harm performance on previously learned abilities. We have revised the writing in Sections 3.2 and 3.3 and removed the misleading PPO-related discussion to avoid confusion.

---

> ### Author Response · Authors · 2025-11-25
>
> **Weakness 5. How to understand the objective of optimization？**
> We fully understand your concern. After removing some misleading content, our final objective is now given in Equation 6. Since we empirically find that low-probability tokens often induce large gradients, our objective focuses on a constrained optimization formulation that selectively stops updates on a subset of tokens in the dataset. Referring to common practices in LLM post-training, we find that clipping is a particularly suitable choice because it operates at the token level, is simple to apply, and is robust in practice.
> Even in PPO, it is non-trivial to formally derive the clipped surrogate objective, as this type of clipped objective is not typical in standard optimization algorithms. Our intention here is not to rely on PPO-style stability arguments, but rather to highlight that this commonly used clipping trick effectively mitigates the large-gradient updates that we identify as a key driver of forgetting.
> Thus, our OPC-SFT objective is to update $\pi_\theta$ in a supervised manner, while simultaneously clipping the updates for off-policy tokens by introducing the policy ratio. In the LLM SFT or imitation-learning setting, weighted supervised learning objectives are widely used [1], [2], [3], and they can be viewed as generalized supervised learning formulations, which also apply to our OPC-SFT objective.
> We also report the performance of importance sampling without clipping below. While it performs reasonably well and achieves strong in-distribution generalization, its performance on several out-of-distribution tasks is poor.
>
> | Method                  | Before RL ALFWorld-seen | Before RL ALFWorld-unseen | After RL ALFWorld-seen | After RL ALFWorld-unseen |
> |-------------------------|-------------------------|---------------------------|------------------------|--------------------------|
> | SFT                     | 75.00                   | 70.90                     | 92.86                  | 89.55                    |
> | Importance Sampling SFT | 75.00                   | 79.10                     | 85.00                  | 85.07                    |
> | OPC-SFT                 | 76.43                   | 77.61                     | 94.29                  | 92.54                    |
>
>
> | Method                  | MBPP  | MMLU  | HumanEval | GPQA  | LiveCodeBench | MATH500 |
> |-------------------------|-------|-------|-----------|-------|---------------|---------|
> | SFT                     | 56.61 | 58.47 | 44.39     | 10.61 | 34.50         | 21.20   |
> | Importance Sampling SFT | 52.65 | 59.20 | 40.01     | 16.67 | 33.85         | 26.20   |
> | OPC-SFT                 | 58.71 | 59.97 | 48.13     | 18.18 | 42.60         | 37.60   |
>
> This result indicates that importance reweighting alone does not change the fact that the process remains supervised learning on a static dataset. It may accelerate optimization, but if certain tokens along this optimization path consistently induce large gradient norms, and consequently lead to forgetting, then stopping those updates is essential.
>
> [1] Token Cleaning: Fine-Grained Data Selection for LLM Supervised Fine-Tuning. ICML 2025.
>
> [2] On the Generalization of SFT: A Reinforcement Learning Perspective with Reward Rectification. CoRR, abs/2508.05629, 2025.
>
> [3] Behavioral Cloning from Noisy Demonstrations. ICLR 2020.

---

> ### Author Response · Authors · 2025-11-25
>
> **Weakness 6 & Question 2 & Question 3. Baselines like replay or decreasing learning rate.**
>
> Thanks for your question. We have added several additional baselines, including KL penalty, decreasing learning rate, and a replay-based continual learning method. For the KL penalty baseline, since computing the exact KL is not efficient, we use the empirical expectation over the offline dataset. For the replay-based baseline, we add a comparable number of math-reasoning samples from DeepScaleR into the SFT dataset. For decreasing learning rate, we reduce the policy model’s learning rate to $1e−6$ during OPC-SFT in the ALFWorld setting. We present the result below.
>
> | Method          | Before RL ALFWorld-seen | Before RL ALFWorld-unseen | After RL ALFWorld-seen | After RL ALFWorld-unseen |
> |-----------------|-------------------------|---------------------------|------------------------|--------------------------|
> | Decreasing LR   | 72.14                   | 66.42                     | 83.57                  | 88.81                    |
> | Replay          | 71.43                   | 77.61                     | 83.57                  | 76.12                    |
> | OPC-SFT         | 76.43                   | 77.61                     | 94.29                  | 92.54                    |
>
>
> | Method          | MBPP  | MMLU  | HumanEval | GPQA  | LiveCodeBench | MATH500 |
> |-----------------|-------|-------|-----------|-------|---------------|---------|
> | Decreasing LR   | 61.11 | 57.03 | 48.24     | 19.70 | 37.94         | 36.40   |
> | Replay          | 55.82 | 61.92 | 46.57     | 16.67 | 38.19         | 46.80   |
> | OPC-SFT         | 58.71 | 59.97 | 48.13     | 18.18 | 42.60         | 37.60   |
>
> Intuitively, KL penalty enforces conservative updates, but it operates at the batch-level. Although it acts as a regularizer on policy updates, our results show that it fails to improve out-of-distribution generalization and even reduces learning efficiency. Decreasing the learning rate slows down the learning of all tokens collectively, but it does not prevent irreversible forgetting. Consequently, the model learns more slowly yet still forgets. Similarly, gradient clipping is a parameter-level constraint that is not specific to individual tokens and also influences the updates for all tokens. The replay-based method also underperforms on the coding tasks, suggesting that designing an effective replay buffer is non-trivial.
>
> In contrast, our method prioritizes tokens that the current policy already handles well, enabling gradual and progressive learning. Tokens that are hard to learn are effectively postponed to later stages in the optimization process, which helps reduce forgetting while maintaining learning efficiency.

---

### Official Review · Reviewer_GBro · 2025-11-06

**Soundness:** 3
**Presentation:** 3
**Contribution:** 3
**Rating:** 8
**Confidence:** 3

**Summary:**

The paper studies the effect of Supervised Finetuning (SFT) procedure in memorization (overfitting) and catastrophic forgetting of the pretraining set. The hypothesis of the authors is that supervised training is dominated by a few tokens that are "off-policy" (i.e that had low likelihood under the current generative model). Fitting these tokens impact most the forgetting, typically during the first epoch.

This probes author to reformulate supervised learning as a special kind of reinforcement learning (in the REINFORCE/PPO-style) with constant advantage A=1. This allows to re-use the clipping trick of PPO on the likelihood ratio between the current policy $\theta$ and $\theta_{old}$.

Multiples experiments are conducted on benchmarks like  ALFworld, LiveCodeBench, or MATH500. Other OOD experiments and ablations can be found in appendix

**Strengths:**

The strength of the method is clearly its simplicity. This is a straightforward modification of existing SFT pipelines.

The numerical results are strong compared to the SFT baseline and the more complex competitor approaches. Three strong backbones of small size (Qwen2.5-7B, Qwen2.5-1.5B and Llama3.2-3B) are finetuned. Their limited capacity make them susceptible to forgetting.

The scientific methodology brings strong evidence of usefulness (e.g Fig. 2, Fig. 3).

Overall, simplicity and extensiveness of experiments, make a strong case for this paper's acceptance.

**Weaknesses:**

### Source of the idea

The idea of reformulating supervised learning as a form of RL has been done in the past. At the end, PPO is just a pretext and a possible interpretation of simply clipping a likelihood ratio. I think it would be better to present it as a form of importance sampling with clipping.
More context on this would be welcome, such as the discussion of the competitor's work [which is acknowledged by authors]

Wenhong Zhu, Ruobing Xie, Rui Wang, Xingwu Sun, Di Wang, and Pengfei Liu. Proximal super-
vised fine-tuning. CoRR, abs/2508.17784, 2025.

### Clipping ratio

Can you report the percentage of clipped ratios as function of time (and epsilon)? I expect this probability to increase until it reaches a plateau (possibly 100%).   It is important to measure it, as every clipped gradient stops bringing signal (wasted token). This is an important measure of the efficiency of the method.  That might have consequences in the data-scarce regime: for very small sets that are OOD, I'm scared that this would stop training completely.

**Questions:**

### Reference policy

Would there be a reason to use anything else than $\pi_0$ as a reference policy $\pi_{\theta}$: e.g, updating it after the end of the first epoch. I'm worried that, by training for too long, the current iterate $\pi$ would become too far from $\pi_{\theta}$, which would trigger clipping repeatedly until no gradient is flowing.

### Baseline

Another baselines comes to mind, related to token probabilities: KL divergence used as regularization between $\pi_{\theta}$ and $\pi_0$. This is regularization approach, whereas PPO can be more understood as a form of constraint like trust region methods. What do you think about this alternative?

---

> ### Author Response · Authors · 2025-11-25
>
> Dear Reviewer GBro,
>
> Thank you for your insightful comments. We sincerely hope that our rebuttal could properly address your concerns.
>
> **Weakness 1. Source of the idea.**
>
> We fully understand your concern. Here we restate the motivation behind our work. We identified that low-probability tokens, which we characterize as off-policy tokens, generate disproportionately large gradient norms, as demonstrated in https://i.postimg.cc/VkJZx1Vz/gradient-bins-two-axes2-01.png, which is our new analysis of the probability distribution versus gradient magnitude. Thus, we aim to clip their updates during the SFT process.
>
> Operationally, our goal is to enforce a token-level constraint that allows updates only for 'proximal' tokens, i.e., those within the trust region. We chose the PPO-style clipping mechanism using the probability ratio
>  $r(\theta) = \frac{\pi_\theta}{\pi_{\theta_{\text{old}}}}$, because it implements this constraint both directly and effectively. With this mechanism, updates driven by low-probability tokens that diverge significantly from the old policy are naturally clipped.
> This ratio clipping acts as a selective filter: it preserves valid learning signals from safe samples while eliminating erratic updates that could lead to catastrophic forgetting. The inherent importance sampling is simply the standard way to deploy this clipping strategy.
>
> We acknowledge that the original text contained redundancy regarding the PPO. Accordingly, we have provided a revised manuscript with a significantly condensed Section 3 to improve clarity.
>
> While our method shares the operational concept of integrating trust regions into SFT with concurrent work [1], our motivation differs significantly. Our approach is driven by the insight that off-policy tokens, characterized by large gradient norms, are the primary drivers of catastrophic forgetting. Accordingly, we focus our evaluation on off-policy scenarios, such as agentic tasks, to demonstrate OPC-SFT's ability to preserve general knowledge. This has been clarified in the Appx.C.1 related work section of our updated manuscript.
>
> [1] Proximal Supervised Fine-Tuning. CoRR, abs/2508.17784, 2025.
>
> **Weakness 2. The percentage of clipped ratio.**
>
> Thank you for your suggestion. We apologize for not clarifying the old policy in the original version; this likely caused a misunderstanding regarding the percentage of clipped tokens. To avoid misunderstanding, we have revised the term 'reference policy' to 'old policy' that updates. As we periodically update the old policy, the number of clipped tokens decreases as the policy updates. Specifically, the number of clipped tokens drops as the update gradient magnitude gradually becomes smaller until convergence, as shown in Figure 1(c).
>
> Eventually, the policy will converge, and the ratio $ \frac{\pi_{\theta}}{\pi_{\theta_{\text{old}}}}$ will approach 1, not exceeding $1 + \epsilon$. We tracked the clipped ratio and found that the number of unclipped tokens increases until it reaches a plateau, where the majority of tokens do not change drastically. We provide the corresponding figure for ALFWorld with $\epsilon = 0.5$ in https://i.postimg.cc/fbgTmGVR/wei-xin-tu-pian-2025-11-24-204851-898.png . If $\epsilon$ is set to a larger value, the number of clipped tokens will decrease further. This analysis has been added in the updated version.

---

> ### Author Response · Authors · 2025-11-25
>
> **Question 1. Reference policy ablation.**
>
> We sincerely apologize for not clearly explaining how the policy $\pi_{\theta_{old}}$ is maintained in our method. To avoid misunderstanding, we have revised the term 'reference policy' to 'old policy' that updates, which is updated after each training step of the policy model on 256 samples. If using the initial policy $\pi_0$ as a fixed $\pi_{\theta_{old}}$, this would indeed cause the issue you pointed out: as training progresses, the current policy could drift too far from $\pi_0$. Instead, we update the old policy periodically during training. Specifically, we reset $\pi_{\theta_{old}}$ after every update batch of 256 samples, which corresponds to one actor training step on 256 trajectories. We refer to this as the old policy update frequency. This ensures that the current iterate does not drift too far from the old policy, preventing excessive clipping and stabilizing the learning dynamics. To further evaluate this design choice, we conducted an ablation study by varying the old policy update frequency, including updating every 128, 512, and 1024 samples. The results show that overly infrequent updates lead to excessive clipping, whereas moderate update frequencies, like 256 samples, achieve stable training. We have also added Sec.4.5 for this ablation study in the updated manuscript.
>
>
> | Old Policy Update Freq | Before RL ALFWorld-seen | Before RL ALFWorld-unseen |
> |------------------------|-------------------------|----------------------------|
> | 128                    | 77.14                   | 76.87                      |
> | 256                    | 78.57                   | 79.10                      |
> | 512                    | 75.00                   | 75.37                      |
> | 1024                   | 73.57                   | 73.88                      |
>
>
> | Old Policy Update Freq | After RL ALFWorld-seen | After RL ALFWorld-unseen |
> |------------------------|------------------------|---------------------------|
> | 128                    | 93.27                  | 91.79                     |
> | 256                    | 94.29                  | 92.54                     |
> | 512                    | 93.57                  | 90.30                     |
> | 1024                   | 89.29                  | 88.81                     |
>
>
> | Old Policy Update Freq | MBPP  | MMLU  | HumanEval | GPQA  | LiveCodeBench | MATH500 |
> |------------------------|-------|-------|-----------|-------|---------------|---------|
> | 128                    | 58.99 | 57.84 | 47.98     | 17.17 | 36.65         | 35.60   |
> | 256                    | 58.71 | 59.97 | 48.13     | 18.18 | 42.60         | 37.60   |
> | 512                    | 61.11 | 57.03 | 48.89     | 19.70 | 36.79         | 36.20   |
> | 1024                   | 60.05 | 59.26 | 46.53     | 21.21 | 35.68         | 36.40   |
>
> **Question 2. More baselines.**
>
> Thanks for the suggestion. We have added the KL penalty baseline to our experiments. As the exact KL cannot be computed efficiently, we utilized the empirical expectation on the offline dataset. Although this acts as a regularizer on policy updates, our results show that it fails to effectively improve out-of-distribution generalization and negatively impacts learning efficiency. Since the KL-Divergence constrains policy updates at the batch-level, it does not accurately clip off-policy tokens. Moreover, the strength of the KL penalty is a hyperparameter that is difficult to set. Due to time limitations, we only evaluated Llama3.2-3B-Instruct on ALFWorld, and additional evaluations will be included in a later version.
>
> | Method      | Before RL ALFWorld-seen | Before RL ALFWorld-unseen | After RL ALFWorld-seen | After RL ALFWorld-unseen |
> |-------------|-------------------------|---------------------------|------------------------|--------------------------|
> | KL penalty  | 72.14                   | 73.13                     | 87.86                  | 82.84                    |
> | OPC-SFT     | 76.43                   | 77.61                     | 94.29                  | 92.54                    |
>
>
>
> | Method      | MBPP  | MMLU  | HumanEval | GPQA  | LiveCodeBench | MATH500 |
> |-------------|-------|-------|-----------|-------|---------------|---------|
> | KL penalty  | 56.61 | 58.12 | 48.70     | 16.16 | 36.83         | 36.20   |
> | OPC-SFT     | 58.71 | 59.97 | 48.13     | 18.18 | 42.60         | 37.60   |

---

### Author Response · Authors · 2025-11-28

# **General Response and Summary of the Rebuttal**

Dear Area Chair,

Thank you for the time and effort you have devoted to evaluating our submission, the reviews, and our rebuttal. We hope this brief summary and general response below will assist in your final assessment.

We sincerely appreciate the recognition of our contributions. The positive points acknowledged by the reviewers are summarized below:

* **Reviewers GBro, tn6h, VFg9** highlighted the simplicity and straightforwardness of our approach, noting that importing a clipping-based gradient strategy offers a simple yet effective solution that is easy to implement.
* **Reviewers N85X, VFg9, GBro** praised the comprehensive diagnostic analyses, recognizing them as strong intermediate evidence supporting the usefulness of our method.
* **Reviewers VFg9, N85X** appreciated the broad empirical validation across tasks and the demonstrated improvements in both in-distribution and out-of-distribution performance.

We also summarize the main clarifications and actions taken to address common concerns. We sincerely thank the reviewers for their efforts and have incorporated their feedback to revise the manuscript for enhanced clarity and impact.

* **Reviewers GBro, tn6h, and VFg9** raised concerns that the connection to PPO appeared artificially constructed. We apologize that our initial presentation caused a misunderstanding, specifically for Reviewer tn6h. We have addressed this by rewriting Section 3 to explain our motivation and remove potentially misleading text clearly.

* **Reviewers tn6h and VFg9** requested stronger evidence for our motivation and questioned whether an intuitive baseline, such as a KL penalty, could suffice. Here we restate the motivation behind our work. We have identified that low-probability tokens, characterized as off-policy tokens, are responsible for large gradient norms, as demonstrated in our newly added analysis https://i.postimg.cc/VkJZx1Vz/gradient-bins-two-axes2-01.png, which shows that large probability shifts strongly correlate with high gradient norms. Therefore, in our method, we aim to clip the updates of these tokens during the SFT process.  To compare with the KL penalty baseline, we have provided complementary experiments in the table below.

Performance on ALFWorld (After RL)
| Model | Method | After RL ALFWorld-seen | After RL ALFWorld-unseen |
| :--- | :--- | :---: | :---: |
| Llama3.2-3B-Instruct | KL penalty | 87.86 | 82.84 |
| | OPC-SFT | **94.29** | **92.54** |
| Qwen2.5-1.5B-Instruct | KL penalty | 86.43 | 83.58 |
| | OPC-SFT | **90.00** | **94.03** |
| Qwen2.5-7B-Instruct | KL penalty | 90.71 | 85.82 |
| | OPC-SFT | **92.14** | **91.04** |


OOD Performance on General Benchmarks (Before RL)
| Model | Method | MBPP | MMLU | HumanEval | GPQA | LiveCodeBench | MATH500 |
| :--- | :--- | :---: | :---: | :---: | :---: | :---: | :---: |
| Llama3.2-3B-Instruct | KL penalty | 56.61 | 58.12 | **48.70** | 16.16 | 36.83 | 36.20 |
| | OPC-SFT | **58.71** | **59.97** | 48.13 | **18.18** | **42.60** | **37.60** |
| Qwen2.5-1.5B-Instruct | KL penalty | 43.92 | 56.63 | 43.64 | 30.30 | **32.94** | 24.00 |
| | OPC-SFT | **46.56** | **58.85** | **44.96** | **33.84** | 32.81 | **24.20** |
| Qwen2.5-7B-Instruct | KL penalty | 75.13 | 67.13 | 74.77 | 33.84 | 56.24 | 70.80 |
| | OPC-SFT | **78.84** | **70.60** | **75.07** | **34.34** | **67.69** | **72.40** |

* **Reviewers N85X, VFg9, GBro** emphasized the need for an ablation study on the old-policy update frequency.
In response, we have included an ablation experiment that varies the update frequency at 128, 256, 512, and 1024 samples on Llama3.2-3B-Instruct, as shown in the tables below.


Performance on ALFWorld (After RL)

| Update Freq | Seen      | Unseen    |
| ----------- | --------- | --------- |
| 128         | 93.27     | 91.79     |
| 256         | **94.29** | **92.54** |
| 512         | 93.57     | 90.30     |
| 1024        | 89.29     | 88.81     |

OOD Performance on General Benchmarks (Before RL)

| Update Freq | MBPP      | MMLU      | HumanEval | GPQA      | LiveCodeBench | MATH500   |
| ----------- | --------- | --------- | --------- | --------- | ------------- | --------- |
| 128         | 58.99     | 57.84     | 47.98     | 17.17     | 36.65         | 35.60     |
| 256         | 58.71     | **59.97** | 48.13     | 18.18     | **42.60**     | **37.60** |
| 512         | **61.11** | 57.03     | **48.89** | 19.70     | 36.79         | 36.20     |
| 1024        | 60.05     | 59.26     | 46.53     | **21.21** | 35.68         | 36.40     |



Due to the limited time, we received no reviewer replies before November 27th. Therefore, we sincerely appreciate the additional time and effort the Area Chair has dedicated to evaluating our submission. We remain committed to refining the paper to more clearly highlight our contributions and improvements.

Best Regards,

ICLR 2026 Conference Submission19335 Authors

---

### Meta-Review · Area_Chair_JTwh · 2026-01-03

**Summary:**

This paper proposes a token-level clipped objective for supervised fine-tuning, aiming to limit extreme per-token updates relative to a reference model and thereby reduce forgetting and produce a more robust cold start for subsequent RL. Reviewers appreciated the simplicity and the reported empirical gains. However, the discussion raised substantial concerns about the method’s positioning and the strength of the evidence for the proposed mechanism. While the rebuttal improves clarity and adds some additional comparisons/ablations, key issues remain insufficiently resolved. Based on the overall discussion, my recommendation is Reject, though I encourage the authors to resubmit after sharpening the contribution boundary and strengthening the evidence.

**Reviewer Concerns:**

Addressed by the rebuttal:

- Clarifies parts of the method description and reduces confusion around how the objective should be interpreted in an SFT setting.

- Adds additional comparisons and some ablations related to the reference model and clipping behavior.

Still outstanding:

- The conceptual motivation remains insufficiently grounded. Several reviewers were not convinced that the proposed objective is meaningfully distinct from established perspectives on token-level weighting or stability regularization, nor that the narrative connection to off-policy RL is necessary or well supported.

- The empirical evidence does not yet isolate the real source of gains. Key analyses needed to separate clipping effects from implicit data reweighting, and to demonstrate robustness across datasets, model sizes, and training recipes, are incomplete.

- Sensitivity to core design choices (clipping threshold, reference refresh schedule, data mixture) is not characterized broadly enough to support strong claims of robustness.

**Reviewer Scores:**

The most positive reviewer: likely unchanged or slightly higher due to added ablations and clarifications.

Reviewers concerned about framing and novelty: likely unchanged, as the main conceptual objections remain.

Reviewers focused on robustness and sensitivity: likely unchanged, since remaining characterization gaps are only partially closed.

---

### Decision · Program_Chairs · 2026-01-26

Reject